# Cascaded Language Models for Cost-Effective Human–AI Decision-Making

**Claudio Fanconi**
University of Cambridge
caf83@cam.ac.uk

**Mihaela van der Schaar**
University of Cambridge
mv472@cam.ac.uk

## Abstract

A challenge in human-AI decision-making is to balance three factors: the *correctness* of predictions, the *cost* of knowledge and reasoning complexity, and the confidence about whether to *abstain* from automated answers or escalate to human experts. In this work, we present a cascaded LLM decision framework that adaptively delegates tasks across multiple tiers of expertise – a base model for initial candidate answers, a more capable and knowledgeable (but costlier) large model, and a human expert for when the model cascade abstains. Our method proceeds in two stages. First, a deferral policy determines whether to accept the base model's answer or regenerate it with the large model based on the confidence score. Second, an abstention policy decides whether the cascade model response is sufficiently certain or requires human intervention. Moreover, to overcome static policies and accommodate changing task difficulty, we incorporate an online learning mechanism which uses human feedback. We demonstrate this approach to general question-answering (ARC-Easy, ARC-Challenge, and MMLU) and medical question-answering (MedQA and MedMCQA). Our results demonstrate that our cascaded strategy outperforms single-model baselines in most cases, achieving higher accuracy while reducing costs and providing a principled approach to handling abstentions.[1]

## 1  Introduction

Data-driven decision support has gained increasing traction in high-stakes fields such as healthcare [Jin et al., 2024, Fan et al., 2024, Li et al., 2024], finance [Li et al., 2023a, Zhao et al., 2024], and education [Xu et al., 2024]. For example, in the medical context, large language models (LLMs) can facilitate accurate diagnoses and treatment recommendations that encode vast knowledge Kim et al. [2024]. However, high accuracy in such complex settings often requires substantial computational resources or multiple reasoning steps. Additionally, LLMs may hallucinate or generate incorrect outputs with severe consequences. Effective human-AI collaboration should balance *correctness*, *cost*, and *abstention*, ensuring AI-driven assistance integrates seamlessly with expert oversight.

**The Challenge.**    A key challenge in effective human–AI collaboration is how to allocate computational and human resources efficiently—deciding when an automated model should answer, when it should escalate to a lager and more capable model, and when it should defer to human expertise. Naïve strategies — such as always relying on the cheaper model or always trusting the more capable one — fail to optimise this trade-off. The former increases the risk of errors and hallucinations, while the latter inflates costs. Likewise, static deferral policies, fixed thresholds, or one-off calibrations cannot adapt to changing task distributions or evolving model competence.

---

[1]We provide the code for our experiments at https://github.com/fanconic/cascaded-llms

39th Conference on Neural Information Processing Systems (NeurIPS 2025).

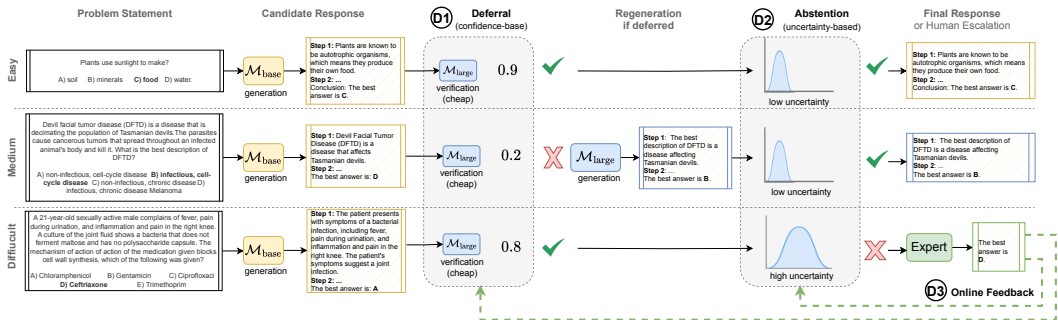

Figure 1: **Cascaded LLM Human-AI Decision-Making Framework Examples.** Given a decision-making problem, the system (1) generates an initial response with a base model, (2) verifies correctness probability, (2.5) defers to a larger model if needed, (3) assesses response uncertainty, and (3.5) abstains to a human expert if necessary. If feedback is available, deferral and abstention modules are adjusted over time. For this system to work efficiently, the modules should uphold three desiderata: (D1) the deferral policy regenerates responses only when necessary, (D2) the abstention policy escalates to humans only when uncertainty is high, (D3) the system continuously improves with feedback.

These limitations motivate three requirements for any cost-effective human–AI decision-making framework:

1. (D1) **Reduce Unnecessary Regenerations:** Responses should only be regenerated by a more capable model when there is sufficient evidence that the current one is unreliable..
2. (D2) **Abstain when Uncertain:** The system should defer to human experts when uncertainty exceeds acceptable bounds, avoiding overconfident automation in high-risk scenarios.
3. (D3) **Adapt over time:** The framework should continuously refine its deferral and abstention policies as feedback becomes available, ensuring sustained reliability and improvement.

Together, these desiderata define the principles an effective decision-making framework must satisfy, irrespective of implementation.

**Our Approach.** We propose a cascaded LLM framework that explicitly satisfies these three requirements. The framework adaptively delegates tasks across multiple tiers of expertise: a lightweight *base model* provides initial answers; a more capable but costlier *large model* regenerates responses when confidence is low; and, if uncertainty remains high in the model-generated responses, the system *abstains* to a human expert. An online learning mechanism continually adjusts the deferral and abstention thresholds based on human feedback, improving decision quality over time. Figure 1 provides an overview of this cascaded decision flow with three example questions of varying difficulty.

**Contributions.** Our main contributions are threefold:

- **Cascaded LLM Human-AI Decision System:** We introduce a multi-tier decision-making system that coordinates LLMs of varying capacity with human experts to balance accuracy, cost, and abstention.
- **Principled Deferral and Abstention Policies:** We design confidence- and uncertainty-based decision policies that regulate when to defer to a larger model or abstain to humans, guided by Bayesian calibration for reliable verification.
- **Online Learning for Adaptive Decision-Making:** We propose an online optimisation scheme that refines the deferral and abstention thresholds using human feedback, enabling continual adaptation to task complexity.

## 2 Related Work

**Multi-LLM Answer Generation.** Several studies have explored collaborative frameworks that leverage multiple LLMs of varying capacities to enhance both performance and cost-efficiency beyond the capabilities of a single model [Chen et al., 2023, Ding et al., 2024, Aggarwal et al., 2024]. Chen et al. [2023] proposed cost-effective strategies such as prompt structuring, model approximation, and cascaded LLM frameworks. Similarly, Ding et al. [2024] introduced an intelligent routing mechanism that dynamically assigns prompts to the most appropriate model. Aggarwal et al. [2024] developed a black-box LLM framework for cost-efficient response generation, formalised as a Partially Observable Markov Decision Process (POMDP), requiring minimal training data. Zhu et al. [2023a] proposed a multiplexer-based approach that balances queries between a small and a large LLM, employing a trained BERT classifier to determine when the smaller model suffices. Šakota et al. [2024] introduced a meta-model-driven selection framework that requires pre-training for optimal query distribution. In a parallel line of research, speculative decoding [Leviathan et al., 2023], employs a lightweight model to generate multiple tokens, which a larger model subsequently verifies.

In contrast to prior research, we propose a multi-tier framework for human-AI collaboration. Rather than relying solely on automation, our approach integrates human intervention when model uncertainty is too high, addressing a gap in previous multi-tier frameworks. Compared to speculative decoding research, our work prioritises the factual correctness of complete responses rather than token-wise distributions, enabling more robust decision-making rather than just fluent text generation. Zellinger et al. [2025] conducts a concurrent line of research that is closest to our work on cascaded LLMs, as well as in their previous works [Zellinger and Thomson, 2024, 2025]. They focus on probabilistic modelling of cascading LLMs and their deferral and abstention mechanisms.

**LLM Answer Verification and Uncertainty Quantification.** Ensuring the reliability of LLM-generated responses requires adequate verification and uncertainty quantification mechanisms. Several studies have explored self-verification strategies [Weng et al., 2023, Jiang et al., 2024, Pan et al., 2024], often leveraging the LLM's internal knowledge [Dhuliawala et al., 2023]. Alternative approaches employ external knowledge sources for verification [Pan et al., 2024, Gao et al., 2023, Peng et al., 2023]. Aggarwal et al. [2024] introduced verification techniques based on available contextual information, predominantly involving multiple LLM queries to validate response accuracy. Another research direction quantifies factual correctness uncertainty [Mahaut et al., 2024]. Kadavath et al. [2022] conducted a detailed analysis of how LLMs express uncertainty through surrogate token probabilities, demonstrating their effectiveness in calibration. Azaria and Mitchell [2023] explored internal LLM states, training classifiers to quantify uncertainty, while methods such as semantic uncertainty estimation [Kuhn et al., 2023] enhance robustness by analysing variations in semantically equivalent token sequences.

Our approach relies on surrogate token probability [Kadavath et al., 2022] as a core verification component. However, we extend this methodology by integrating a hierarchical escalation mechanism that dynamically transitions between models and human experts based on verification results.

**Selective Prediction.** Selective prediction enables models to abstain from uncertain queries [El-Yaniv and Wiener, 2010], a crucial feature in risk-sensitive settings where errors are costly. The idea dates back to Chow's work on optical character recognition [Chow, 1957, 1970], and has since been shown to improve deep learning performance [Geifman and El-Yaniv, 2017]. In NLP, abstention has been introduced through confidence-based thresholds [Xin et al., 2021, Yoshikawa and Okazaki, 2023], with recent work on uncertainty quantification for large language models advancing this line of research [Manakul et al., 2023, Farquhar et al., 2024, Lin et al., 2024].

**LLMs in Online Learning.** Traditional LLM research predominantly evaluates language models on static datasets. However, our work aligns with online learning paradigms, wherein policies are continuously refined in response to streaming data [Cortes et al., 2018, Ye et al., 2024]. Our methodology is inspired by Jarrett et al. [2022], who introduced an online decision mediation framework mediating between suboptimal human decisions and an expert oracle. A similar research with the online learning approach is conducted by Zhu et al. [2023a], which extended their multiplexer mechanism to an online setting.

## 3 Background

### 3.1 Cascaded Decision System

We consider a two-tiered cascaded LLM decision system for question answering under resource constraints, denoted by $C = \mathcal{M}_{\text{base}} \to \mathcal{M}_{\text{large}}$, following the notation of Zellinger et al. [2025]. Let $x \in \mathcal{X}$ be a problem statement or prompt, and let $y \in \mathcal{Y}$ denote a system-generated response. For every input $x$, the models return a confidence score $\Phi_i(x) \in [0, 1]$ and an uncertainty score $\Xi_i(x) \in [0, \infty)$, where $i \in \{\text{base}, \text{large}\}$. The decision to predict using the base model $\mathcal{M}_{\text{base}}$ or to defer to the larger model $\mathcal{M}_{\text{large}}$ is based on whether the confidence exceeds a deferral threshold, i.e., $\Phi(x) > \phi_{\text{base}}$. Thus, a prediction is only made if the base model is sufficiently confident. In contrast, abstention is governed by predictive uncertainty: if this exceeds a threshold, $\Xi_i(x) > \xi_i$, the system abstains and forwards the query to a human expert.

We formally define the cascaded decision system as:

$$C(x) = \begin{cases} \mathcal{M}_{\text{base}}(x) & \text{if } \Phi_{\text{base}}(x) > \phi_{\text{base}} \wedge \Xi_{\text{base}}(x) < \xi_{\text{base}} \\ \mathcal{M}_{\text{large}}(x) & \text{if } \Phi_{\text{base}}(x) \leq \phi_{\text{base}} \wedge \Xi_{\text{base}}(x) \leq \xi_{\text{base}} \wedge \Xi_{\text{large}}(x) \leq \xi_{\text{large}} \\ \varnothing & \text{if } \Xi_{\text{base}}(x) \geq \xi_{\text{base}} \vee \Xi_{\text{base}}(x) \geq \xi_{\text{base}} \end{cases} \tag{1}$$

The decision flow of this cascade is also illustrated in Figure 2. While we focus on a two-model system here, the framework naturally generalises to cascades involving multiple LLMs of varying sizes.

The objective is to generate accurate responses while accounting for the computational costs of the models and abstaining when the system is too uncertain. As described in Zellinger and Thomson [2024], this constitutes a multi-objective optimisation problem over three dimensions: error, cost, and abstention. Formally, we minimise the system risk:

$$\mathcal{R}(C) = \mathbb{P}(\text{error} \wedge \neg\text{abstention}) + \lambda_c \mathbb{E}[\text{Cost}] + \lambda_a \mathbb{P}(\text{abstention}) \tag{2}$$

Here, $\mathbb{P}(\text{error} \wedge \neg\text{abstention})$ denotes the probability of the system making an error when it does not abstain, $\mathbb{E}[\text{Cost}]$ is the expected computational cost incurred, and $\mathbb{P}(\text{abstention})$ is the probability of the system abstaining and deferring to a human expert. The terms $\lambda_c$ and $\lambda_a$ weight the cost and abstention penalties, respectively. We explain the system risk in more detail in Section 4.2.

**Assumptions.** (A1) The base model is cost-efficient but less accurate, whereas the large model is more capable but computationally expensive. (A2) Generating responses incurs significantly higher cost than processing inputs, especially in settings that require Chain-of-Thought (CoT) prompting [Wei et al., 2022] or advanced test-time reasoning [Xie et al., 2024]. (A3) Each response is assumed to be either correct or incorrect, with no ambiguity.

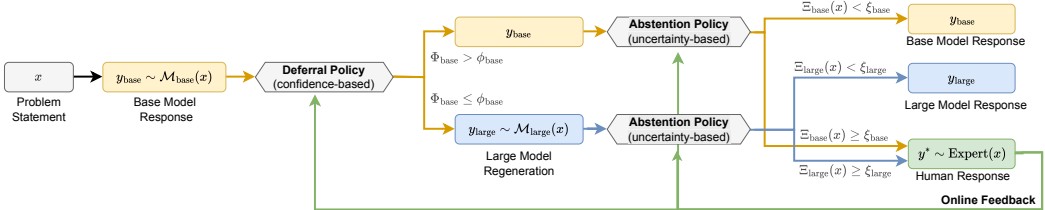

Figure 2: **Decision flow of the two-tiered cascaded LLM system.** The base model first evaluates each query. Confident, low-uncertainty responses are accepted; uncertain ones are passed to the large model or, if still uncertain, deferred to a human expert. Online feedback progressively improves these policies.

### 3.2 Cost Calculation

To estimate the computational cost of response generation, we define a cost function that scales linearly with model size and token counts. Let $s$ denote the model size (in billions of parameters,

e.g., Llama3.1-8B $\Rightarrow s = 8$), $t_{\text{in}}$ and $t_{\text{out}}$ the numbers of input and output tokens, and $\rho > 0$ the output-to-input token cost ratio accounting for the higher cost of generation. The total cost is given by:

$$\text{Cost}(s, t_{\text{in}}, t_{\text{out}}, \rho) = s \cdot (t_{\text{in}} + \rho \cdot t_{\text{out}}) \tag{3}$$

This provides a simple yet effective way to compare models of different sizes under a unified cost metric, independent of infrastructure-specific pricing. Additional cost components can be incorporated as needed.

## 4 Methods

### 4.1 Calibrated Confidence and Uncertainty Estimation

Effective deferral and abstention decisions in a cascaded system critically depend on accurately quantifying model confidence $\Phi(x)$ and uncertainty $\Xi(x)$ for each input $x$. Overconfident or miscalibrated predictions can lead to errors, while excessive uncertainty may result in unnecessary escalations. Therefore, the first part of our method focuses on analysing a range of techniques to estimate these quantities in a reliable and cost-efficient manner. To this end, we evaluate four complementary methods that approximate the probability that a response is correct.

**(1) Self-Verification.** Given an input $x$ and a model response $y_{\text{base}} \sim \mathcal{M}_i(x)$, we prompt the same model to quantify how likely the response is correct by generating a new response [Li et al., 2023b]. The model returns raw confidence score by outputting either a scalar value token in response to a verification prompt (see Appendix B.2). The outputted probability serves as an uncalibrated estimate of correctness.

**(2) Consistent Self-Verification.** We repeat the self-verification process $n$ times under stochastic sampling (e.g., with temperature), and aggregate the resulting probabilities. The empirical mean forms the uncalibrated confidence score. This approach is inspired by self-consistency as in [Aggarwal et al., 2024].

**(3) Surrogate Token Probability.** We adopt the approach of Kadavath et al. [2022], where the model $\mathcal{M}_i$ is asked to verify whether a generated response $y$ is correct, and we extract the next-token probability over the discrete label set YES/NO. Specifically:

$$p_i(x) = \frac{\mathcal{M}_i(\text{YES} \mid x, y)}{\mathcal{M}_i(\text{YES} \mid x, y) + \mathcal{M}_i(\text{NO} \mid x, y)}, \tag{4}$$

**(4) Monte-Carlo Surrogate Token Probability.** To obtain better confidence estimates, we apply Monte Carlo Dropout [Gal and Ghahramani] at test time when computing the surrogate token probability. For each of $n$ stochastic forward passes, we sample an estimate $\hat{p}_i^{(t)}(x)$, and the empirical mean forms the uncalibrated confidence score:

$$p_i(x) = \frac{1}{T} \sum_{t=1}^{n} \hat{p}_i^{(t)}(x) \tag{5}$$

**Model Evaluation by Larger Models.** For each of the above methods, the evaluating model $\mathcal{M}_i$ can either be the same model that generated the original response, or a larger model in the cascade, if available. While self-evaluation is cheap and self-contained, verifying a small model's output using a larger model is still substantially cheaper than generating a new response from scratch—particularly when generation involves long-form reasoning, as per Assumption A2. Additionally, larger models tend to be better calibrated and may yield more reliable verification, improving downstream deferral and abstention decisions [Zhu et al., 2023b, Chhikara, 2025].

**Bayesian Calibration.** To ensure that the extracted confidence scores are comparable across models and consistent with empirical correctness, we fit a Bayesian logistic regression model on a small calibration set of 100 samples. This is a Bayesian version of Platt scaling [Platt, 2000], and we assume a Normal distribution as prior. We follow Zellinger and Thomson [2024]'s approach and

apply a non-linear transformation on the raw confidence score before inputting it into the Bayesian model, to spread out the clusters of overconfident probabilities.

$$p_{tr}(p_i) = \begin{cases} \log(\frac{1}{1-p_i}) & \text{if } p_i \geq 0.5 \\ \log(2) - \log(\frac{1}{p_i}) & \text{if } p_i < 0.5 \end{cases} \tag{6}$$

Subsequently, the Bayesian Logistic Regression outputs a posterior distribution over correctness. The mean of the posterior predictive distribution defines the calibrated confidence $\Phi(x)$, while we use standard deviation as a model-based uncertainty estimate $\Xi(x)$, as in [Fanconi et al., 2023].

## 4.2 Online Improvement

To enable online learning (D3), we parameterise the deferral and abstention thresholds and optimise them online. Given a dataset $\mathcal{D}^{(t)}$ at time $t \in \mathbb{N}$ with previous problem statements and ground truth labels, we update the thresholds using stochastic gradient descent. While the system is deployed, we assume that we will receive a ground truth response $(y^*)$ at the end of every decision if the system abstains. Thus, our dataset continually increases $\mathcal{D}^{(t)} = \mathcal{D}^{(t-1)} \cup \{x, y^*\}$ every time the cascade abstains.

Our objective function is the system risk $\mathcal{R}(C)$ (Equation 2). We expand this risk into the concrete, differentiable losses. Throughout, let

$$\Phi_i(x) \in [0, 1], \qquad \Xi_i(x) \in [0, 1], \qquad i \in \{\text{base}, \text{large}\}$$

denote the *calibrated probability of correctness* (posterior predictive) and a *uncertainty score* (i.e. posterior predictive standard deviation) returned by model $i$ for an input $x$. The optimisation variables are

$$\phi_{\text{base}}, \ \xi_{\text{base}}, \ \xi_{\text{large}} \in (0, 1),$$

For numerical stability we treat their raw, unconstrained versions $\phi_{\text{base}}^{\text{raw}}, \xi_{\text{base}}^{\text{raw}}, \xi_{\text{large}}^{\text{raw}} \in \mathbb{R}$ as the true optimisation parameters and map them to $(0, 1)$ with a sigmoid function:

$$\phi_{\text{base}} = \sigma(\phi_{\text{base}}^{\text{raw}}), \quad \xi_{\text{base}} = \sigma(\tau_{\text{base}}^{\text{raw}}), \quad \xi_{\text{large}} = \sigma(\tau_{\text{large}}^{\text{raw}}).$$

To keep the loss fully differentiable, we replace every Boolean test with a soft logistic step, where $k$ determines the steepness

$$\mathbf{1}\{z > 0\} \ \longrightarrow \ g_k(z) = \sigma(k\,z).$$

With this convention the three mutually exclusive masks at the *base* stage are

$$p_{\text{abst1}}(x) = g_k\big(\Xi_{\text{base}}(x) - \xi_{\text{base}}\big), \tag{7}$$

$$m_{\text{pred1}}(x) = \big(1 - m_{\text{abst1}}\big) \cdot g_k\big(\Phi_{\text{base}}(x) - \phi_{\text{base}}\big), \tag{8}$$

$$m_{\text{defer1}}(x) = \big(1 - m_{\text{abst1}}\big) \cdot g_k\big(\phi_{\text{base}} - \Phi_{\text{base}}(x)\big), \tag{9}$$

and the masks at the *large* stage are

$$p_{\text{abst2}}(x) = m_{\text{defer1}}(x) \cdot g_k\big(\Xi_{\text{large}}(x) - \xi_{\text{large}}\big), \tag{10}$$

$$m_{\text{pred2}}(x) = m_{\text{defer1}}(x) \cdot \big(1 - g_k(\Xi_{\text{large}}(x) - \xi_{\text{large}})\big). \tag{11}$$

**Probability of abstention.** The cascade abstains in two mutually exclusive ways, so

$$\mathbb{P}(\text{abstention}) \ = \ p_{\text{abst1}} + p_{\text{abst2}}. \tag{12}$$

**Expected correctness.** Only the *prediction* masks contribute a non-zero probability of correctness; we weight each by the calibrated confidence:

$$\mathbb{E}[\text{Correct}] \ = \ \mathbb{E}\big[m_{\text{pred1}} \cdot \Phi_{\text{base}}\big] \ + \ \mathbb{E}\big[m_{\text{pred2}} \cdot \Phi_{\text{large}}\big]. \tag{13}$$

**Expected cost.** Let $c_1$ be the costs from the base model, which consist of the generation cost and the verification cost (either by itself or by a larger model). Furthermore, $c_2$ is the generation cost and the verification cost caused by the large model. The first term is incurred on every query; the second is incurred only if we defer:

$$\mathbb{E}[\text{Cost}] \ = \ c_1 \ + \ \mathbb{E}[m_{\text{defer1}}] \cdot c_2. \tag{14}$$

**System-risk objective.** Substituting the three expectations above into Eq. (2) produces the differentiable loss that is back-propagated during threshold optimisation in online learning:

$$\mathcal{R}(C) = 1 - \mathbb{E}[\text{Correct}] \ + \ \lambda_c \, \mathbb{E}[\text{Cost}] \ + \ \lambda_a \big(p_{\text{abst1}} + p_{\text{abst2}}\big). \tag{15}$$

# 5  Experiments

In this section, we empirically assess whether the desiderata (D1), (D2), and (D3), introduced in Section 1, are satisfied. For (D1) and (D2), we analyse in Section 5.1 the performance of various confidence estimation techniques with respect to calibration and cost-efficiency. Subsequently, in Section 5.2, we investigate whether the system improves through online learning.

**General Setup.** We evaluate a cascade of two LLMs, specifically (Qwen-2.5-1.5B → Qwen-2.5-7B). Additional results for other cascades—(Llama3.2-3B → Llama3.1-8B), (Llama3.2-1B → Llama3.1-8B), and (Qwen-2.5-3B → Qwen-2.5-7B)—are reported in Appendix C. These model pairs are selected due to their open-source availability and our ability to run them on an NVIDIA A100 GPU.

To evaluate the generalisability of our framework across domains, we use five question-answering datasets: (1) ARC2-Easy and (2) ARC2-Challenge [Clark et al., 2018], which are part of the AI2 Reasoning Challenge and require reasoning over grade-school science; (3) Massive Multitask Language Understanding (MMLU) benchmark [Hendrycks et al., 2021], which covers 57 subjects ranging from complex STEM to international law, nutrition, and religion; and two medical QA benchmarks: (4) MedQA [Jin et al., 2020], consisting of US medical board exam questions, and (5) MedMCQA [Pal et al., 2022], comprising entrance exam questions from the Indian medical school curriculum. All datasets are in multiple-choice format, with ground-truth answers satisfying Assumption (A3). Chain-of-Thought reasoning is employed to generate answers. The cost proportion between input and output tokens is set to $\rho = 5$, consistent with Anthropic's current pricing to date [Anthropic, 2025]. Details on generation and verification prompts can be found in Appendix B.2.

## 5.1  Cost-Benefit Analysis of Verification Methods

We begin by empirically analysing which verification method from Section 4.1 is most suitable for estimating the confidence of a generated response. Once calibrated via Bayesian logistic regression, these confidence estimates determine whether to defer a prediction from the base model to the larger model.

To assess both cost-efficiency and accuracy, we compare the calibrated base model confidence $\Phi_{\text{base}}$ against two baselines: (1) using only the base model (Qwen-2.5-1.5B) and (2) using only the large model (Qwen-2.5-7B). In Figure 3, we visualise accuracy versus cost per sample across the datasets. We use a threshold-agnostic strategy where deferral to the large model is performed with probability $\Phi_{\text{base}}(x)$. We evaluate four methods: Self-Verification (SV, $n=1$), Surrogate Token Probability (STP, $n=1$), Consistent Self-Verification (SV, $n=5$), and Monte Carlo STP (MC-STP, $n=5$). For the latter two, we perform five regenerations or stochastic passes. Each experiment is conducted once using $\mathcal{M}_{\text{base}}$ as the verifying LLM, and once using $\mathcal{M}_{\text{large}}$.

As shown in Figure 3 (and Figures 7, 10, and 13 in Appendix C), using a larger model for verification generally yields a better cost-benefit profile, particularly on simpler datasets (ARC2-Easy, ARC2-Challenge, MMLU). In contrast, base-model verification provides only marginal gains. On the more complex medical datasets (MedQA and MedMCQA), all methods struggle. STP ($n=1$) is the most effective the ARC2-Easy and ARC2-Challenge dataset.

To quantitatively assess cost-efficiency, we compute the Incremental Benefit per Cost (IBC) metric from Aggarwal et al. [2024], defined as:

$$\text{IBC}_{\text{cascade}} = \frac{P_{\text{cascade}} - P_{\text{base}}}{C_{\phi_{\text{m2m}}} - C_{\text{base}}}, \quad \text{IBC}_{\text{base}} = \frac{P_{\text{large}} - P_{\text{base}}}{C_{\text{large}} - C_{\text{base}}},$$

where $P$ denotes accuracy and $C$ denotes cost. We then compute the relative gain:

$$\Delta\text{IBC} = \frac{\text{IBC}_{\text{cascade}} - \text{IBC}_{\text{base}}}{\text{IBC}_{\text{base}}} \cdot 100.$$

Higher $\Delta\text{IBC}$ values indicate improved cost-efficiency over the baseline.

As seen in Table 1 verifying with $\mathcal{M}_{\text{large}}$ consistently leads to higher $\Delta\text{IBC}$ scores, particularly on ARC2-Easy, ARC2-Challenge, and MMLU. On the medical datasets, no single method consistently

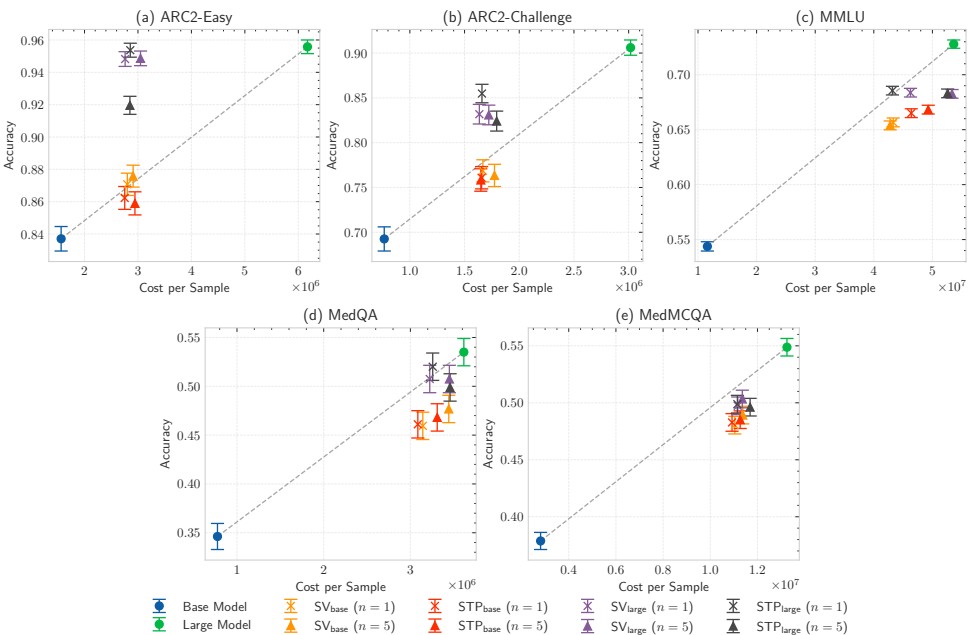

Figure 3: **Cost-Accuracy Trade-off for Calibrated Verification Methods (Qwen-2.5 1.5B→7B).**
Accuracy versus cost per sample is shown for the cascaded model using various verification methods.
Performance above the linear interpolation line between base and large model baselines indicates a
positive cost-benefit. Error bars represent standard error.

|  |  | ARC2 Easy | ARC2 Challenge | MMLU | MedQA | MedMCQA |
|---|---|---|---|---|---|---|
| **Base** | SV ($n$=1) | -2.4 $\pm$ 32.6 | -11.6 $\pm$ 20.5 | -18.7 $\pm$ 4.9 | -27.3 $\pm$ 14.3 | -24.8 $\pm$ 9.5 |
| | SV ($n$=5) | 10.2 $\pm$ 29.6 | -26.2 $\pm$ 19.1 | -19.4 $\pm$ 4.9 | -25.4 $\pm$ 13.4 | -19.5 $\pm$ 9.4 |
| | STP ($n$=1) | -16.9 $\pm$ 33.2 | -19.3 $\pm$ 20.9 | -20.4 $\pm$ 4.5 | -23.9 $\pm$ 14.7 | -22.9 $\pm$ 9.7 |
| | MC-STP ($n$=5) | -37.4 $\pm$ 28.8 | -16.0 $\pm$ 21.0 | -24.6 $\pm$ 4.2 | -27.3 $\pm$ 13.7 | -22.2 $\pm$ 9.4 |
| **Large** | SV ($n$=1) | **258.4** $\pm$ 39.1 | 70.5 $\pm$ 26.0 | -7.8 $\pm$ 4.7 | -6.1 $\pm$ 15.8 | -11.6 $\pm$ 10.0 |
| | SV ($n$=5) | 188.5 $\pm$ 31.5 | 51.0 $\pm$ 23.7 | -24.1 $\pm$ 3.9 | -11.9 $\pm$ 14.7 | -11.0 $\pm$ 9.8 |
| | STP ($n$=1) | 242.7 $\pm$ 36.3 | **89.3** $\pm$ 26.2 | **2.5** $\pm$ 5.2 | **1.7** $\pm$ 16.3 | **-10.0** $\pm$ 10.1 |
| | MC-STP ($n$=5) | 171.4 $\pm$ 36.4 | 39.2 $\pm$ 22.0 | -22.4 $\pm$ 4.0 | -16.8 $\pm$ 14.5 | -19.7 $\pm$ 9.3 |

Table 1: **Calibrated $\Delta$IBC Scores for Qwen-2.5 (1.5B→7B).** Each row indicates a verification
method (SV or STP) with $n = 1$ or $n = 5$, grouped by whether the base or large model was used for
verification.

outperforms the others significantly. Moreover, we see that on the medical datasets, the $\Delta$IBC
standard error rates for the verification scores using the large model are around 0, indicating no
cost-benefit compared to the easier datasets. We report additional results for the other cascades in
Tables 2, 3, 4, 5, and 6 in Appendix C. Interestingly, the uncalibrated confidence scores appear to
yield higher $\Delta$IBC, albeit with a significantly higher standard error, suggesting the instability of
uncalibrated confidence scores. We conduct an ablation study of the size of the calibration set in
Appendix C.5, which demonstrates that the calibration size between 50-500 samples, does not lead to
a significant performance change. Moreover, we report on the various subjects of the MMLU dataset
in Appendix Section C.6, which reveals a stark difference in $\Delta$IBC scores across different areas of
expertise.

### 5.2 Online Improvement of the Decision System

Desideratum **D3** requires that "the framework should continuously refine its deferral and abstention
policies as feedback becomes available, ensuring sustained reliability and improvement". We simulate

an online setting in which the system selects among $\mathcal{M}_{\text{base}}$, $\mathcal{M}_{\text{large}}$, or a human expert, adjusting its thresholds based on feedback from abstentions.

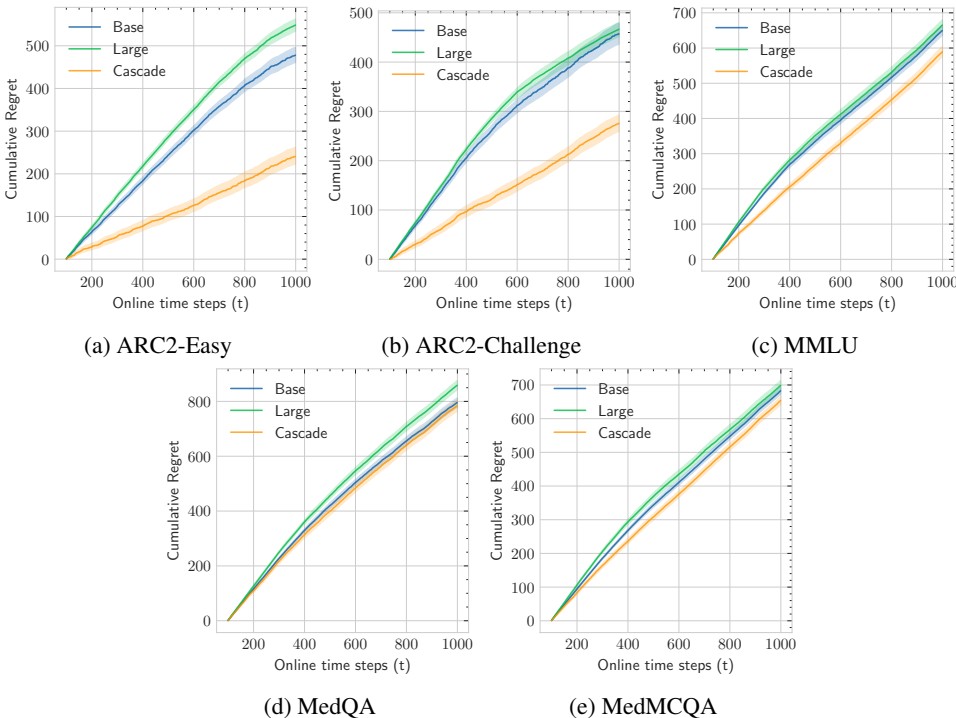

(a) ARC2-Easy         (b) ARC2-Challenge         (c) MMLU

(d) MedQA         (e) MedMCQA

Figure 4: **Cumulative Regret in Online Setting (Qwen-2.5 1.5B → 7B).** Cumulative system risk over time. Training data is collected only when abstentions occur. The cascaded system consistently achieves lower regret.

Specifically, the experiment streams 1,000 unseen questions in a random order to simulate production traffic. Before the first query, the deferral probabilities are calibrated using the values learned from the 100-sample calibration set. Thereafter, we add queries that were marked for abstention and answered by an oracle expert to a replay buffer. This replay buffer is used to perform the ADAM optimiser [Kingma and Ba, 2014] updates on the differentiable risk 2 with a learning rate of 0.05 and a batch size of 10, on the deferral and abstention thresholds $\boldsymbol{\theta} = \{\phi_{\text{base}}, \xi_{\text{base}}, \xi_{\text{large}}\}$. The prediction is made on an unseen query and the regret is calculated on it. If a query is added to the replay buffer, the regret associated with it has already been calculated, and the sample now becomes a training sample; however, it is not further evaluated. We compare the cascaded system $C$ to using only $\mathcal{M}_{\text{base}}$ and only $\mathcal{M}_{\text{large}}$ with a single abstention threshold $\xi$. The system risk for a single model is explained in more detail in Appendix A.1. As we are considering a deployed system, we track the cumulative regret over time, which we define as follows:

$$\mathbf{Regret}(\mathcal{M})[n] \coloneqq \sum_{t=1}^{n} \mathcal{R}(\mathcal{M}^{(t)}),$$

where $\mathcal{M} \in \{C, \mathcal{M}_{\text{base}}, \mathcal{M}_{\text{large}}\}$ and $\mathcal{M}^{(t)}$ evolves based on abstention feedback $\mathcal{D}_t$. We chose regret as the metric for this experiment, inspired by the work on online decision mediation [Jarrett et al., 2022]. Regret is the running sum of our per-query risk. Because error, compute, and human-hand-off are already weighted into the same units, adding them over time tells you the exact "bill" the system has paid. A lower regret curve indicates a higher benefit, as it represents a combination of abstentions, correct predictions, and costs, and we can see it grow over time in a deployed setting. The regret curve illustrates how quickly a policy learns online and whether early mistakes are compensated for later.

We initialise the thresholds at $\boldsymbol{\theta}^{(0)} = \{0.5, 0.05, 0.05\}$, where $\xi_i = 0.05$ corresponds to the standard deviation of 5% confidence. For the single model baselines, we initialise the abstention threshold

as with $\xi = 0.05$, and keep the rest of the hyperparameters the same. Throughout this experiment, we employ the STP ($n = 1$) verification strategy, which was found to be the most competitive in the previous section. To avoid trivial solutions (e.g., always selecting one model), we balance system risk using $\lambda_c = 10^{-5}$ and $\lambda_a = 0.1$, in line with Zellinger et al. [2025].

Figure 4 shows that the cascaded system yields lower cumulative regret over 1000 test samples on ARC2-Easy, ARC2-Challenge, MMLU, and MedMCQA, compared to using either model in isolation. On MedQA, gains are less clear, likely due to poor confidence estimation, which was also observed in the section above. Similar trends are observed in other cascades (see Figures 9, 12, 15 in Appendix C). Nevertheless, in four of the five cases, the cascaded LLM system demonstrates lower cumulative regret than when using single models online, where feedback is received when abstaining from action. Additionally, we experiment by comparing our proposed gradient-based approach to a traditional grid search over $\theta$ in Appendix C.7.

## 5.3 The Effect of Imperfect Expert

To recall the system risk objective in Equation 15, the part where wrong human annotations will have an impact is the expected correctness (Equation 13). More precisely, the calibrated confidence scores $\Phi_{\text{base}}$ and $\Phi_{\text{large}}$. The noisier the feedback is, the more uncalibrated $\Phi_{\text{base}}$ and $\Phi_{\text{large}}$ will become. Therefore, optimisation of the cascaded model will become unreliable.

We demonstrate this through an additional experiments in the online setting, where we progressively swap correct to incorrect predictions while calibrating the model, and how this affects the trajectory on the ARC-Easy dataset with `Qwen-2.5-1.5B → Qwen-2.5-7B`. The results are displayed in Figure 5. We observe that the higher the percentage of label corruption is in the calibration set, the higher the cumulative regret becomes while deploying the decision-making system.

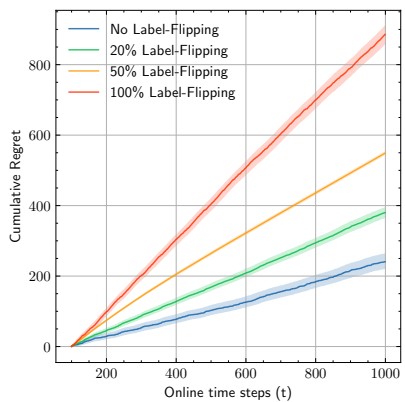

Figure 5: **Imperfect Experts**. We increase the percentage of flipped labels during system calibration, simulating imperfect experts, which in turn increases the system's risk of error.

## 6 Limitations

Our cascaded multi-LLM decision-making framework strikes a balance between accuracy, cost, and abstention, but it has limitations. Sensitivity to cost and abstention variations can impact efficiency, leading to trivial solutions (only using the cheapest model or the model with the lowest error rate). Discrepancies in model performance or relative costs may lead to over-reliance on specific models, thereby reducing adaptability. Furthermore, parameter initialisation affects the convergence of the deferral policy. Additionally, the framework relies on human feedback, which may hinder adaptation if it is sparse or noisy in a real-world scenario. Finally, fitting a Bayesian logistic regression model is usually more complex than fitting a regular one, depending on the different posterior approximations or sampling strategies employed.

## 7 Conclusion

We proposed a multi-tier decision-making framework that escalates tasks between a base model, a large model, and human experts. By leveraging deferral and abstention policies, our approach aims to enhance performance, accuracy, and abstention while adapting through online learning. Our experiments show that the framework outperforms single-model baselines by reducing unnecessary escalations and improving response correctness on the ARC2-Easy, ARC2-Challenge, MMLU, and MedMCQA datasets. On MedQA, a cascaded model did not outperform the single model approach, potentially due to the complexity of the dataset. Nevertheless, we believe that this proposed system could be beneficial where performance, costs, and abstention of LLMs need to be carefully balanced. Future work should investigate different uncertainty quantification methods of LLMs to enhance abstention. Moreover, it would be crucial to examine whether there are theoretical guarantees that justify the application of cascaded LLMs.

## Acknowledgements and Disclosure of Funding

We want to extend our gratitude to Alan Jeffares, Paulius Rauba, Yusuke Kano, Jeremy Voisey, and Alison Smithard for their insightful discussions and valuable feedback. Canon Medical Systems Corporation funds CF's studentship. This work was supported by Microsoft's Accelerate Foundation Models Academic Research initiative.

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

# A  Additional Method Details

## A.1  Single-model System Risk

For completeness we report the risk of running either model *alone*. If $\xi$ is that model's abstention threshold and $c$ its total cost,

$$\mathcal{R}_{\text{single}}(\mathcal{M}) = 1 - \mathbb{E}\big[(1 - m_{\text{abst}})\Phi\big] + \lambda_c c + \lambda_a \mathbb{E}[m_{\text{abst}}],$$

where $m_{\text{abst}}(x) = g_k\big(\Xi(x) - \xi\big)$.

# B  Implementation Details

The code for this paper, to reproduce the results is provided at https://github.com/fanconic/cascaded-llms. All experiments are implemented in Python [Van Rossum and Drake Jr, 1995] with Py-Torch [Paszke et al., 2017] and Hugging Face Transformers [Wolf et al., 2020].

**Compute.**  Experiments are conducted on a single A100-class GPUs.

## B.1  Generation Models

Policies are initialised from instruction-tuned checkpoints and trained with the learned reward signal. The following policy backbones are used:

- `meta-llama/Llama-3.2-1B-Instruct`
- `meta-llama/Llama-3.2-3B-Instruct`
- `meta-llama/Llama-3.1-8B-Instruct`
- `Qwen/Qwen2.5-1.5B-Instruct`
- `Qwen/Qwen2.5-3B-Instruct`
- `Qwen/Qwen2.5-7B-Instruct`

## B.2  Prompts

Throughout this paper, we use prompts to make decision predictions using Chain-of-Thought and verification prompts to determine a response's factual correctness or uncertainty.

---

**Response Generation Prompt ARC2-Easy + ARC2-Challenge**

```
You are a helpful AI.
Answer the following multiple-choice question using step-by-step reasoning,
then conclude with a final line stating the best answer.

Question: {question}

Choices:
{choice_0}
{choice_1}
{choice_2}
{choice_3}
({choice_4})

Let's reason step-by-step, then conclude with: "The best answer is: <X>"

Reasoning:
```

---

**Response Generation Prompt MMLU**

```
You are an expert in {subject}.
Answer the following multiple-choice question using step-by-step reasoning,
then conclude with a final line stating the best answer.

Question: {question}

Choices:
{choices}

Let's reason step-by-step, then conclude with: "The best answer is: <X>"

Reasoning:
"""
```

**Response Generation Prompt MedQA**

```
You are a medical doctor taking the US Medical Licensing Examination.
Answer the following multiple-choice question using step-by-step reasoning,
then conclude with a final line stating the best answer.

Question: {question}

Choices:
{choice_0}
{choice_1}
{choice_2}
{choice_3}
{choice_4}

Let's reason step-by-step, then conclude with: "The best answer is: <X>"

Reasoning:
```

**Response Generation Prompt MedMCQA**

```
You are a medical doctor answering real world medical entrance exam questions.
Answer the following multiple-choice question using step-by-step reasoning,
then conclude with a final line stating the best answer.

Question: {question}

Choices:
{choice_0}
{choice_1}
{choice_2}
{choice_3}

Let's reason step-by-step, then conclude with: "The best answer is: <X>"

Reasoning:
```

**Self Verification Prompt**

```
Given the following question and the model's answer, please evaluate correctness.
Question: {question}

Model Answer: {candidate_answer}

Please give a confidence score on a scale of 0.0 to 1.0 for this prediction.

Answer:
```

**Surrogate Token Probability Prompt**

```
Given the following question and the model's answer, please evaluate correctness.
Respond with a single token: {yes_token} or {no_token}

Question: {question}

Model Answer: {candidate_answer}

Is this answer correct: {yes_token} or {no_token}?

Answer:
```

# C Additional Results

## C.1 Qwen-2.5 1.5B → 7B

### C.1.1 Uncalibrated

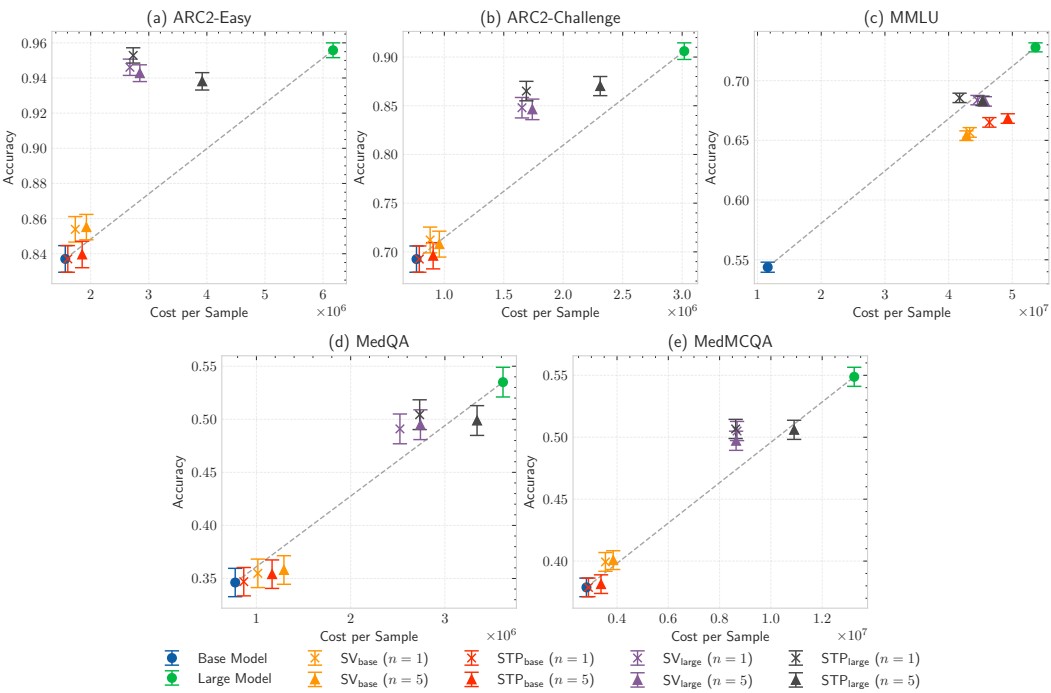

Figure 6: **Benefit-Cost Analysis of *Uncalibrated* Verification Methods *(Qwen-2.5 1.5B-7B)*.** We display the cost vs accuracy of the various verification methods, using the cascade (`Qwen-2.5-1.5B` → `Qwen-2.5-7B`). Verification methods, which are located above the linear interpolation between the base or large models, indicate a positive cost-benefit ratio. The error bars indicate the standard error.

| | | ARC2 Easy | ARC2 Challenge | MMLU | MedQA | MedMCQA |
|---|---|---|---|---|---|---|
| **Base** | SV ($n$=1) | $278.5 \pm 237.3$ | $80.5 \pm 174.1$ | $-18.7 \pm 4.9$ | $-45.8 \pm 118.8$ | $70.4 \pm 89.0$ |
| | SV ($n$=5) | $93.3 \pm 112.7$ | $-15.6 \pm 104.2$ | $-19.4 \pm 4.9$ | $-65.6 \pm 55.3$ | $29.3 \pm 63.2$ |
| | STP ($n$=1) | $-100.0 \pm 948.0$ | $-100.0 \pm 847.4$ | $-20.4 \pm 4.5$ | $-87.0 \pm 313.1$ | $-100.0 \pm 835.5$ |
| | MC-STP ($n$=5) | $-66.3 \pm 142.6$ | $-74.3 \pm 143.4$ | $-24.6 \pm 4.2$ | $-69.7 \pm 72.9$ | $-71.6 \pm 114.7$ |
| **Large** | SV ($n$=1) | $280.5 \pm 41.6$ | $84.8 \pm 24.6$ | $-3.0 \pm 5.0$ | $\mathbf{24.7} \pm 21.0$ | $32.0 \pm 14.0$ |
| | SV ($n$=5) | $219.7 \pm 35.8$ | $66.5 \pm 22.3$ | $-7.0 \pm 4.8$ | $13.9 \pm 18.9$ | $24.6 \pm 13.8$ |
| | STP ($n$=1) | $\mathbf{284.5} \pm 40.4$ | $\mathbf{96.9} \pm 24.2$ | $\mathbf{7.5} \pm 5.5$ | $21.6 \pm 19.4$ | $\mathbf{35.0} \pm 14.2$ |
| | MC-STP ($n$=5) | $66.5 \pm 19.2$ | $21.3 \pm 14.6$ | $-5.7 \pm 4.9$ | $-10.5 \pm 14.6$ | $-3.5 \pm 10.2$ |

Table 2: *Uncalibrated* $\Delta$**IBC Scores for Qwen-2.5 (1.5B→7B).** Each row indicates a verification method (SV or STP) with $n = 1$ or $n = 5$, grouped by whether the base or large model was used for verification.

## C.2 LLama3 1B → 8B

### C.2.1 Calibrated

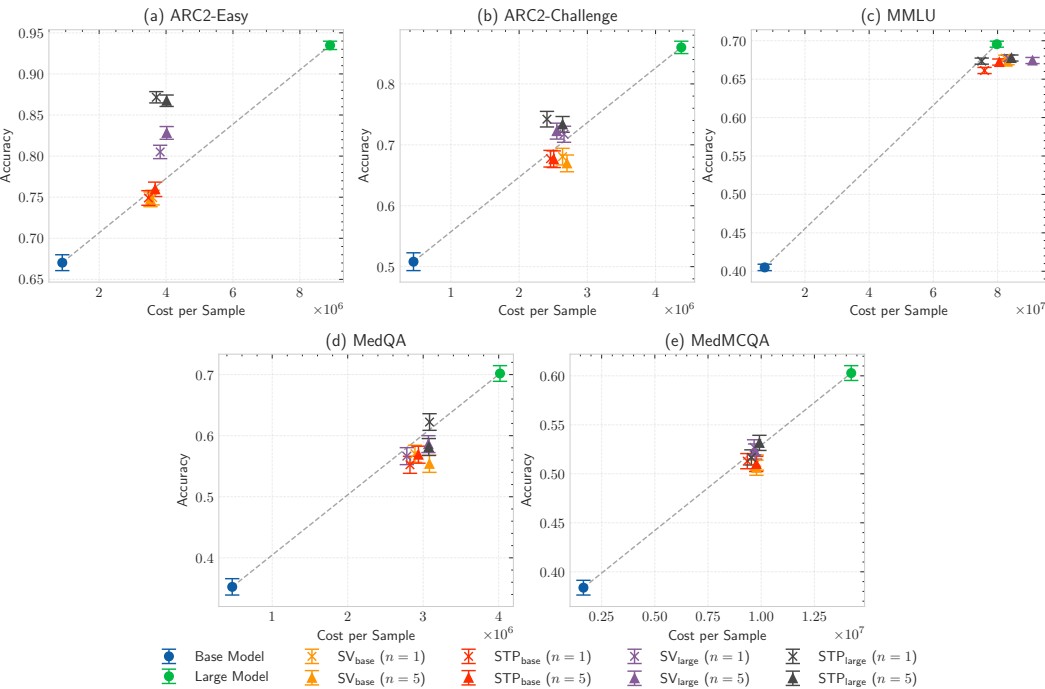

Figure 7: **Benefit-Cost Analysis of *Calibrated* Verification Methods *(Llama3 1B→8B)*.** We display the cost vs accuracy of the various verification methods, using the cascade (`Llama3.2-1B` → `Llama3.1-8B`). Verification methods, which are located above the linear interpolation between the base or large models, indicate a positive cost-benefit ratio. The error bars indicate the standard error.

|        |              | ARC2 Easy      | ARC2 Challenge | MMLU          | MedQA          | MedMCQA        |
|--------|--------------|----------------|----------------|---------------|----------------|----------------|
| **Base**  | SV ($n$=1)    | -8.8 ± 16.4    | -13.2 ± 11.4   | -9.6 ± 2.6    | -8.0 ± 9.5     | -9.9 ± 8.9     |
|        | SV ($n$=5)    | -12.8 ± 16.5   | -19.6 ± 10.8   | -11.9 ± 2.6   | -21.3 ± 8.6    | -15.6 ± 8.7    |
|        | STP ($n$=1)   | -4.4 ± 16.7    | -5.2 ± 11.7    | -6.8 ± 2.8    | -13.6 ± 9.3    | -4.8 ± 9.3     |
|        | MC-STP ($n$=5)| 0.8 ± 15.2     | -9.0 ± 11.4    | -9.0 ± 2.7    | -9.4 ± 9.2     | -11.2 ± 8.8    |
| **Large** | SV ($n$=1)    | 38.6 ± 14.5    | 2.8 ± 11.1     | -10.6 ± 2.6   | -4.1 ± 9.9     | -0.0 ± 9.1     |
|        | SV ($n$=5)    | 50.1 ± 13.7    | 10.7 ± 11.8    | -19.7 ± 2.3   | -8.7 ± 9.1     | -3.1 ± 9.0     |
|        | STP ($n$=1)   | **118.3** ± 15.8 | **38.1** ± 13.7 | **-1.2** ± 2.9 | **5.6** ± 9.7  | -3.7 ± 9.2     |
|        | MC-STP ($n$=5)| 97.7 ± 14.4    | 15.4 ± 11.7    | -11.7 ± 2.5   | -6.8 ± 9.2     | **2.6** ± 9.0  |

Table 3: *Calibrated* ΔIBC Scores for Llama3 (1B→8B). Each row indicates a verification method (SV or STP) with $n = 1$ or $n = 5$, grouped by whether the base or large model was used for verification.

### C.2.2 Uncalibrated

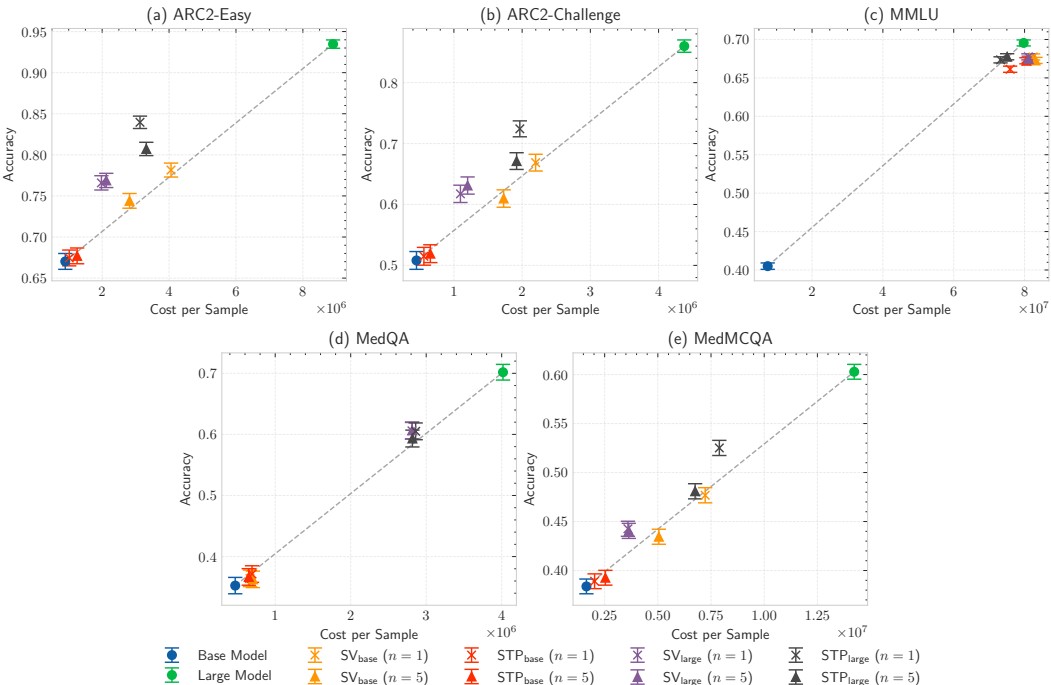

Figure 8: **Benefit-Cost Analysis of *Uncalibrated* Verification Methods *(Llama3 1B→8B)*.** We display the cost vs accuracy of the various verification methods, using the cascade (`Llama3.2-1B →` `Llama3.1-8B`). Verification methods, which are located above the linear interpolation between the base or large models, indicate a positive cost-benefit ratio. The error bars indicate the standard error.

|  |  | **ARC2 Easy** | **ARC2 Challenge** | **MMLU** | **MedQA** | **MedMCQA** |
|---|---|---|---|---|---|---|
| **Base** | SV ($n$=1) | $6.5 \pm 13.1$ | $2.6 \pm 13.8$ | $-9.6 \pm 2.6$ | $-38.9 \pm 98.5$ | $-4.2 \pm 12.0$ |
|  | SV ($n$=5) | $16.1 \pm 21.3$ | $-11.3 \pm 18.4$ | $-11.9 \pm 2.6$ | $-55.7 \pm 82.4$ | $-14.4 \pm 18.6$ |
|  | STP ($n$=1) | $6.4 \pm 344.3$ | $-30.0 \pm 211.7$ | $-6.8 \pm 2.8$ | $-13.9 \pm 87.0$ | $-21.2 \pm 159.6$ |
|  | MC-STP ($n$=5) | $-42.5 \pm 116.2$ | $-39.0 \pm 113.6$ | $-9.0 \pm 2.7$ | $-19.8 \pm 107.9$ | $-42.9 \pm 68.9$ |
| **Large** | SV ($n$=1) | $\mathbf{167.9} \pm 38.0$ | $\mathbf{89.3} \pm 36.6$ | $-7.9 \pm 2.7$ | $9.9 \pm 10.2$ | $\mathbf{73.3} \pm 32.8$ |
|  | SV ($n$=5) | $143.4 \pm 33.6$ | $83.1 \pm 31.6$ | $-9.1 \pm 2.6$ | $\mathbf{10.1} \pm 10.2$ | $63.5 \pm 32.0$ |
|  | STP ($n$=1) | $129.4 \pm 19.1$ | $59.1 \pm 16.5$ | $\mathbf{1.6} \pm 3.0$ | $7.3 \pm 9.9$ | $30.0 \pm 11.8$ |
|  | MC-STP ($n$=5) | $71.0 \pm 17.2$ | $24.0 \pm 16.5$ | $0.3 \pm 2.9$ | $4.2 \pm 10.0$ | $9.2 \pm 13.2$ |

Table 4: *Uncalibrated* $\Delta$**IBC Scores for Llama3 (1B→8B).** Each row indicates a verification method (SV or STP) with $n = 1$ or $n = 5$, grouped by whether the base or large model was used for verification.

### C.2.3 Online Learning

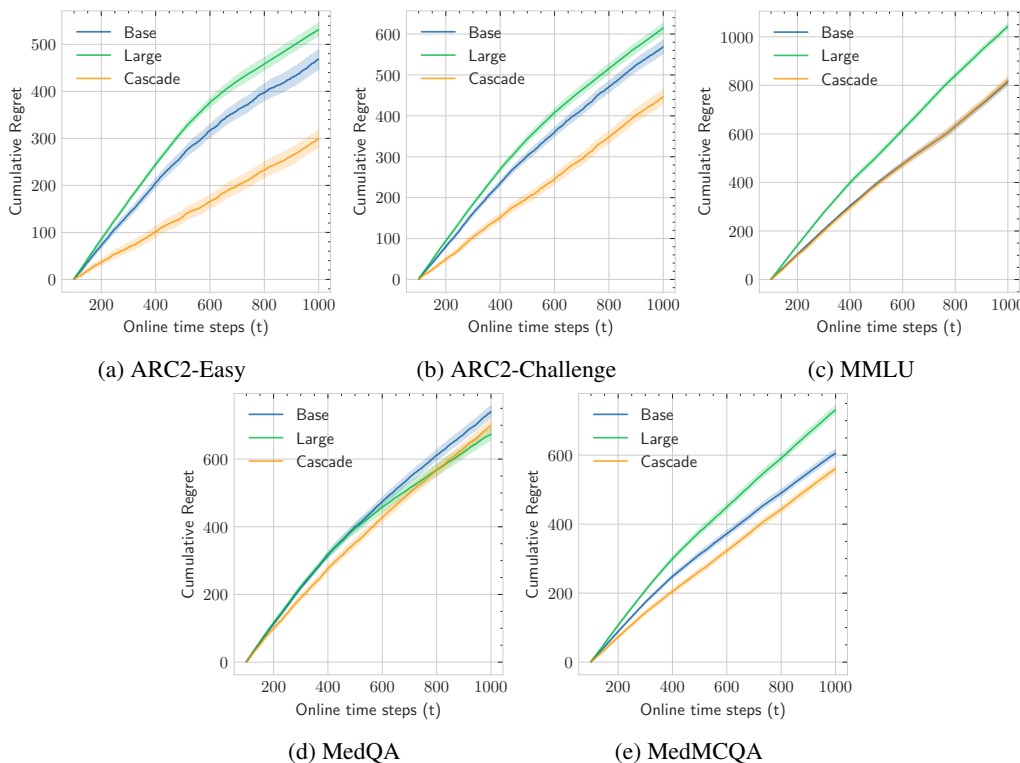

(a) ARC2-Easy        (b) ARC2-Challenge        (c) MMLU

(d) MedQA        (e) MedMCQA

Figure 9: **Cumulative Regret in Online Setting** *(Llama3 1B→8B)*. We display the cumulative regret of the system risk when using Cascade (`Llama3.2-1B` → `Llama3.1-8B`). Points are only added to the training set if an abstention is made. The error bars indicate the standard error.

### C.3  LLama 3B → 8B

Due to the large size of the MMLU dataset and the costs associated with making predictions on it, we omit this for the (Llama3.2-3B → Llama3.1-8B) combination in this subsection.

| | | ARC2 Easy | | ARC2 Challenge | | MedQA | | MedMCQA | |
|---|---|---|---|---|---|---|---|---|---|
| | | *uncal.* | *cal.* | *uncal.* | *cal.* | *uncal.* | *cal.* | *uncal.* | *cal.* |
| **Base** | SV ($n=1$) | $267.7 \pm 364.0$ | $3.8 \pm 96.2$ | $322.3 \pm 335.2$ | $-11.5 \pm 81.4$ | $15.7 \pm 205.7$ | $-48.6 \pm 46.9$ | $93.6 \pm 154.5$ | $-27.8 \pm 20.9$ |
| | SV ($n=5$) | $247.2 \pm 359.8$ | $32.1 \pm 100.9$ | $352.2 \pm 325.3$ | $0.0 \pm 84.4$ | $12.5 \pm 200.0$ | $-28.2 \pm 49.4$ | $96.1 \pm 146.7$ | $-39.5 \pm 20.4$ |
| | STP ($n=1$) | $272.7 \pm 338.6$ | $34.9 \pm 99.6$ | $279.7 \pm 337.5$ | $-40.7 \pm 80.9$ | $-35.6 \pm 199.1$ | $\mathbf{-8.2} \pm 49.3$ | $91.7 \pm 153.0$ | $-19.2 \pm 21.9$ |
| | MC-STP ($n=5$) | $272.4 \pm 325.0$ | $44.0 \pm 96.7$ | $279.9 \pm 312.6$ | $33.0 \pm 100.0$ | $-9.2 \pm 204.9$ | $-23.2 \pm 52.8$ | $\mathbf{119.0} \pm 154.3$ | $-45.0 \pm 20.0$ |
| **Large** | SV ($n=1$) | $312.2 \pm 321.9$ | $\mathbf{84.2} \pm 110.6$ | $237.4 \pm 277.6$ | $-20.5 \pm 84.8$ | $-12.4 \pm 136.7$ | $-25.3 \pm 49.1$ | $81.0 \pm 100.1$ | $-23.3 \pm 21.5$ |
| | SV ($n=5$) | $257.4 \pm 324.7$ | $63.2 \pm 106.4$ | $281.6 \pm 292.7$ | $\mathbf{37.4} \pm 83.1$ | $4.7 \pm 131.5$ | $-32.4 \pm 48.8$ | $84.8 \pm 107.0$ | $-14.2 \pm 22.1$ |
| | STP ($n=1$) | $331.9 \pm 287.7$ | $77.0 \pm 112.2$ | $254.6 \pm 291.8$ | $-0.7 \pm 83.7$ | $-8.8 \pm 126.8$ | $-34.4 \pm 46.2$ | $95.7 \pm 99.5$ | $-38.6 \pm 20.3$ |
| | MC-STP ($n=5$) | $\mathbf{422.2} \pm 328.8$ | $78.2 \pm 109.8$ | $\mathbf{325.5} \pm 306.1$ | $-6.6 \pm 78.7$ | $\mathbf{65.2} \pm 150.7$ | $-52.8 \pm 48.1$ | $91.4 \pm 100.3$ | $\mathbf{-11.5} \pm 22.1$ |

Table 5: ΔIBC scores for **Llama3 (3B→8B)** across datasets and calibration settings. Rows show methods (SV or STP) with $n = 1$ or $n = 5$, grouped by whether probabilities come from the base or large model. All values are rounded to 1 decimal place.

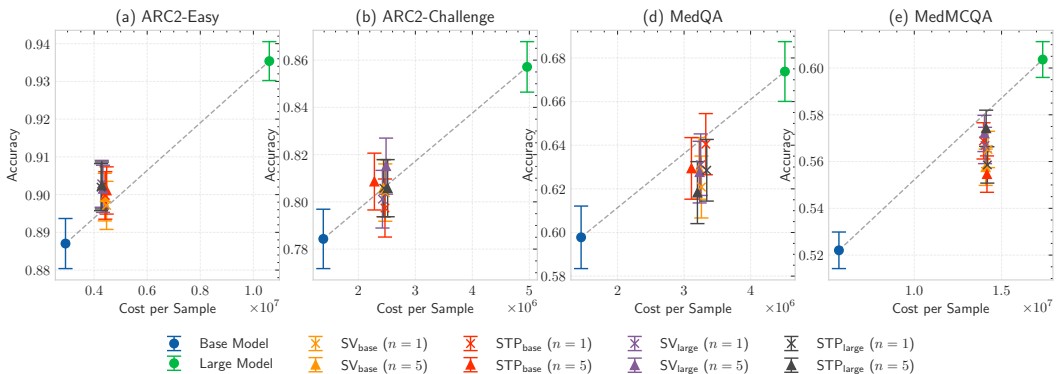

Figure 10: **Benefit-Cost Analysis of *Calibrated* Verification Methods *(Llama3 3B→8B)*.** We display the cost vs accuracy of the various verification methods, using the cascade (Llama3.2-3B → Llama3.1-8B). Verification methods, which are located above the linear interpolation between the base or large models, indicate a positive cost-benefit ratio. The error bars indicate the standard error.

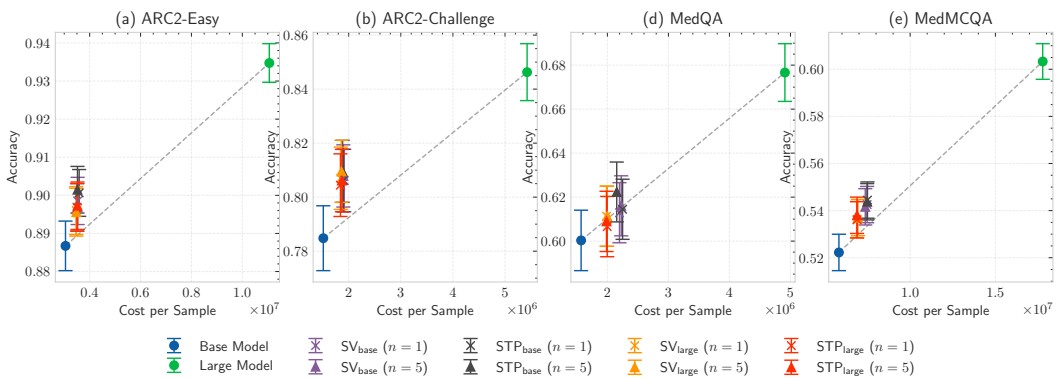

Figure 11: **Benefit-Cost Analysis of *Uncalibrated* Verification Methods *(Llama3 3B→8B)*.** We display the cost vs accuracy of the various verification methods, using the cascade (Llama3.2-3B → Llama3.1-8B). Verification methods, which are located above the linear interpolation between the base or large models, indicate a positive cost-benefit ratio. The error bars indicate the standard error.

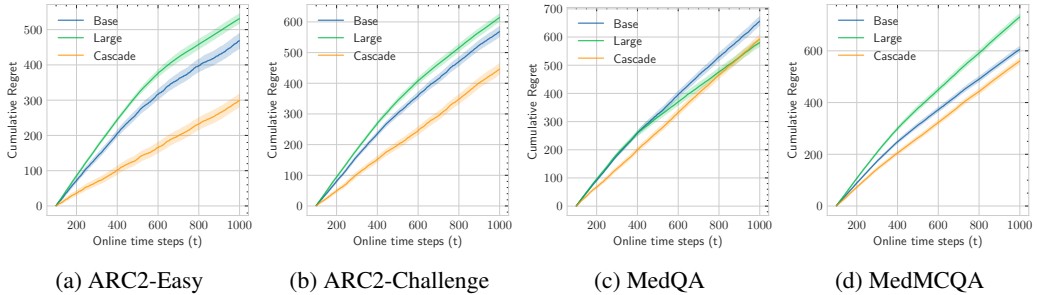

(a) ARC2-Easy  (b) ARC2-Challenge  (c) MedQA  (d) MedMCQA

Figure 12: **Cumulative Regret in Online Setting** *(Llama3 3B→8B)*. We display the cumulative regret of the system risk when using Cascade (`Llama3.2-3B` → `Llama3.1-8B`). Points are only added to the training set if an abstention is made. The error bars indicate the standard error.

## C.4 Qwen 3B → 7B

Due to the large size of the MMLU dataset and the costs associated with making predictions on it, we omit this for the (`Qwen-2.5-3B` → `Qwen-2.5-7B`) combination in this subsection.

|  |  | **ARC2 Easy** | | **ARC2 Challenge** | | **MedQA** | | **MedMCQA** | |
|---|---|---|---|---|---|---|---|---|---|
|  |  | *uncal.* | *cal.* | *uncal.* | *cal.* | *uncal.* | *cal.* | *uncal.* | *cal.* |
| **Base** | SV ($n=1$) | $72.6 \pm 67.1$ | $80.6 \pm 98.5$ | $-35.1 \pm 45.8$ | $-3.9 \pm 62.5$ | $-37.5 \pm 27.8$ | $-46.2 \pm 15.7$ | $-21.0 \pm 21.1$ | $-40.3 \pm 14.4$ |
|  | SV ($n=5$) | $3.7 \pm 48.0$ | $13.7 \pm 76.4$ | $-17.8 \pm 39.0$ | $-59.1 \pm 50.3$ | $-51.3 \pm 20.8$ | $-54.6 \pm 13.2$ | $-39.9 \pm 19.0$ | $-46.2 \pm 13.5$ |
|  | STP ($n=1$) | $-19.4 \pm 23.5$ | $44.7 \pm 97.3$ | $-42.1 \pm 16.8$ | $-53.6 \pm 57.0$ | $-41.5 \pm 12.1$ | $-52.0 \pm 15.5$ | $-37.5 \pm 10.6$ | $-31.5 \pm 15.3$ |
|  | MC-STP ($n=5$) | $-32.1 \pm 21.8$ | $-7.5 \pm 75.5$ | $-43.3 \pm 18.1$ | $-53.2 \pm 53.8$ | $-47.8 \pm 11.9$ | $-50.4 \pm 13.7$ | $-39.9 \pm 11.0$ | $-45.5 \pm 14.0$ |
| **Large** | SV ($n=1$) | $246.6 \pm 106.2$ | $278.9 \pm 125.3$ | $83.6 \pm 56.8$ | $43.7 \pm 65.6$ | $-9.8 \pm 20.5$ | $-46.3 \pm 15.4$ | $10.0 \pm 21.6$ | $-29.6 \pm 15.6$ |
|  | SV ($n=5$) | $99.0 \pm 60.4$ | $139.1 \pm 84.2$ | $21.5 \pm 38.7$ | $27.4 \pm 58.1$ | $-34.9 \pm 14.2$ | $-43.0 \pm 14.2$ | $-13.4 \pm 17.7$ | $-30.3 \pm 14.9$ |
|  | STP ($n=1$) | $342.9 \pm 121.3$ | $313.9 \pm 135.6$ | $96.1 \pm 57.9$ | $99.4 \pm 75.8$ | $-8.3 \pm 20.0$ | $-23.7 \pm 17.4$ | $16.3 \pm 22.2$ | $-17.1 \pm 16.5$ |
|  | MC-STP ($n=5$) | $21.0 \pm 35.6$ | $97.2 \pm 89.8$ | $-19.4 \pm 24.9$ | $16.2 \pm 56.8$ | $-52.8 \pm 11.1$ | $-44.7 \pm 14.2$ | $-37.4 \pm 12.9$ | $-29.5 \pm 14.9$ |

Table 6: $\Delta$IBC scores for **Qwen-2.5 (3B→7B)** across datasets and calibration settings. Rows show methods (SV or STP) with $n = 1$ or $n = 5$, grouped by whether probabilities come from the base or large model. All values are rounded to 1 decimal place.

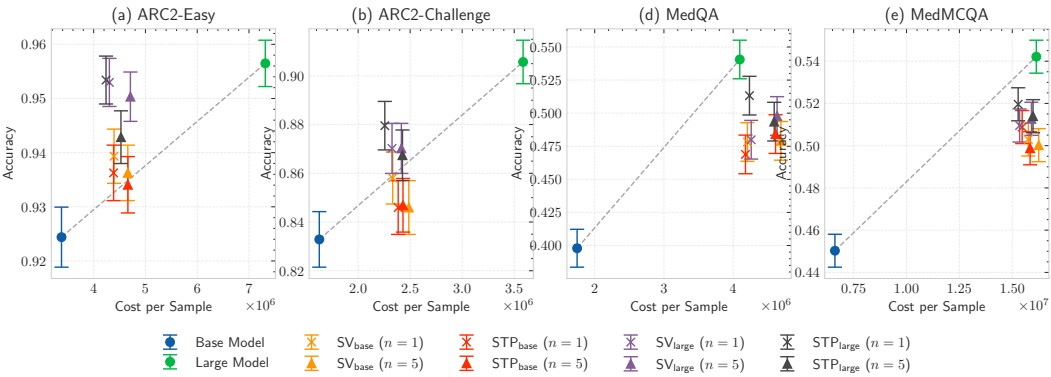

Figure 13: **Benefit-Cost Analysis of** *Calibrated* **Verification Methods** *(Qwen-2.5 3B→7B)*. We display the cost vs accuracy of the various verification methods, using the cascade (`Qwen-2.5-3B` → `Qwen-2.5-7B`). Verification methods, which are located above the linear interpolation between the base or large models, indicate a positive cost-benefit ratio. The error bars indicate the standard error.

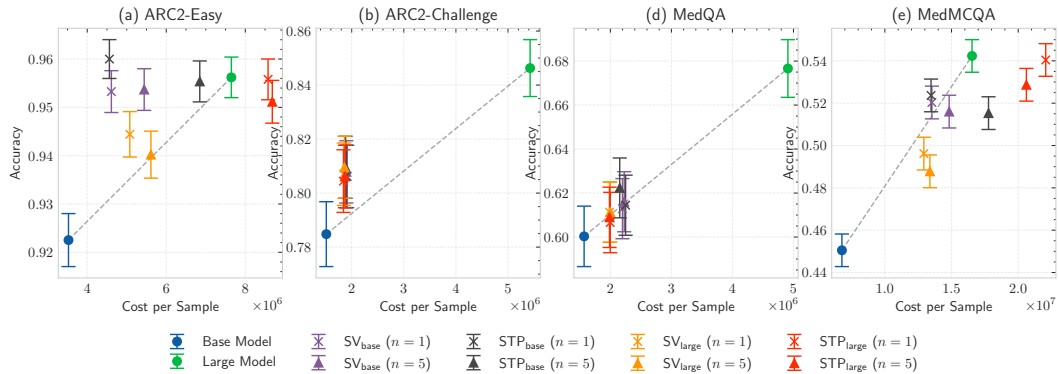

Figure 14: **Benefit-Cost Analysis of *Uncalibrated* Verification Methods *(Qwen-2.5 3B→7B)*.** We display the cost vs accuracy of the various verification methods, using the cascade (Qwen-2.5-3B → Qwen-2.5-7B). Verification methods, which are located above the linear interpolation between the base or large models, indicate a positive cost-benefit ratio. The error bars indicate the standard error.

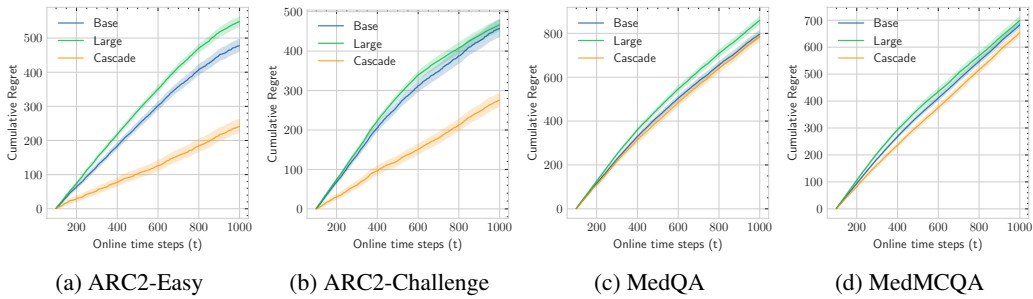

| (a) ARC2-Easy | (b) ARC2-Challenge | (c) MedQA | (d) MedMCQA |

Figure 15: **Cumulative Regret in Online Setting *(Qwen-2.5 3B→7B)*.** We display the cumulative regret of the system risk when using Cascade (Qwen-2.5-3B → Qwen-2.5-7B). Points are only added to the training set if an abstention is made. The error bars indicate the standard error.

## C.5   Ablation of Different Calibration Size

We conducted an ablation study to investigate the effect of calibration size on deferral probability verification, using STP ($n = 1$). The results can be found here in Table 7.

Generally, it appears that the calibration size has little influence on these magnitudes. If we examine the standard error across the various sizes, none of them is significantly better than the others. The only thing that we noted was that with a too small calibration set, we would have more diverging chains in the Bayesian Logistic Regression sampling.

| Model | Dataset | Cal. Size = 50 | Cal. Size = 100 | Cal. Size = 200 | Cal. Size = 500 |
|-------|---------|----------------|-----------------|-----------------|-----------------|
| Qwen-2.5 (1.5B→7B) | ARC2 Easy | $239.5 \pm 36.1$ | $242.7 \pm 36.3$ | $\mathbf{273.5} \pm 42.7$ | $254.9 \pm 43.5$ |
| | ARC2 Challenge | $87.8 \pm 24.3$ | $\mathbf{89.3} \pm 26.2$ | $68.7 \pm 25.4$ | $62.1 \pm 28.7$ |
| | MedQA | $-3.0 \pm 15.4$ | $\mathbf{1.7} \pm 16.3$ | $-7.7 \pm 16.6$ | $-16.6 \pm 18.6$ |
| | MedMCQA | $-7.5 \pm 9.9$ | $-10.0 \pm 10.1$ | $-8.0 \pm 9.9$ | $\mathbf{-5.7} \pm 10.4$ |
| Llama 3 (1B→8B) | ARC2 Easy | $121.9 \pm 15.6$ | $118.3 \pm 15.8$ | $\mathbf{122.6} \pm 16.5$ | $102.8 \pm 16.0$ |
| | ARC2 Challenge | $\mathbf{45.7} \pm 13.1$ | $38.1 \pm 13.7$ | $40.3 \pm 12.8$ | $43.4 \pm 15.6$ |
| | MedQA | $\mathbf{10.1} \pm 9.7$ | $5.6 \pm 9.7$ | $5.3 \pm 10.0$ | $9.8 \pm 11.9$ |
| | MedMCQA | $\mathbf{7.4} \pm 9.5$ | $-3.7 \pm 9.2$ | $6.7 \pm 9.1$ | $7.0 \pm 9.3$ |

Table 7: $\Delta$IBC scores across different calibration sizes for Qwen-2.5 and Llama-3 models on multiple datasets, using STP ($n = 1$) as verification strategy.

## C.6 MMLU Subject

| Subject | ΔIBC (Base → Large) |
| --- | --- |
| International Law | 58.66 ± 89.93 |
| US Foreign Policy | 54.20 ± 147.34 |
| Jurisprudence | 45.95 ± 76.81 |
| Business Ethics | 33.53 ± 77.09 |
| Sociology | 33.10 ± 66.72 |
| High School Psychology | 31.38 ± 37.80 |
| High School Government And Politics | 29.64 ± 35.23 |
| Logical Fallacies | 28.68 ± 87.21 |
| World Religions | 23.14 ± 68.51 |
| Human Aging | 20.88 ± 47.71 |
| Philosophy | 18.97 ± 44.90 |
| Computer Security | 16.84 ± 440.64 |
| Miscellaneous | 16.78 ± 20.86 |
| Management | 16.30 ± 61.84 |
| High School Microeconomics | 15.60 ± 27.56 |
| High School Geography | 13.66 ± 29.49 |
| Marketing | 12.23 ± 61.58 |
| Prehistory | 11.94 ± 33.99 |
| High School Biology | 11.23 ± 42.87 |
| Security Studies | 10.47 ± 50.74 |
| Medical Genetics | 8.21 ± 34.18 |
| College Biology | 8.20 ± 39.82 |
| Professional Psychology | 4.43 ± 22.52 |
| High School US History | 4.28 ± 30.17 |
| Clinical Knowledge | 3.93 ± 48.41 |
| Formal Logic | 3.25 ± 46.67 |
| Human Sexuality | 3.07 ± 65.82 |
| All Subjects (Average) | 2.52 ± 5.23 |
| Anatomy | 2.32 ± 54.35 |
| Public Relations | 1.82 ± 152.51 |
| College Medicine | 1.19 ± 46.04 |
| Global Facts | 0.70 ± 51.04 |
| High School Macroeconomics | 0.52 ± 26.42 |
| High School European History | 0.24 ± 65.82 |
| Abstract Algebra | − |
| Nutrition | -1.55 ± 28.62 |
| Professional Accounting | -1.67 ± 31.01 |
| High School Chemistry | -1.85 ± 26.87 |
| High School Mathematics | -2.46 ± 29.03 |
| Machine Learning | -2.84 ± 40.49 |
| Moral Disputes | -3.36 ± 36.88 |
| Elementary Mathematics | -3.60 ± 23.81 |
| Conceptual Physics | -3.78 ± 29.27 |
| High School Computer Science | -4.81 ± 44.25 |
| Professional Law | -6.02 ± 21.39 |
| High School Physics | -6.83 ± 21.16 |
| College Physics | -8.02 ± 28.96 |
| Astronomy | -8.60 ± 30.55 |
| High School Statistics | -10.18 ± 26.16 |
| Econometrics | -10.50 ± 36.09 |
| Electrical Engineering | -10.79 ± 41.71 |
| College Mathematics | -12.30 ± 62.39 |
| High School World History | -16.96 ± 53.38 |
| Professional Medicine | -18.75 ± 21.12 |
| Moral Scenarios | -22.65 ± 16.20 |
| College Chemistry | -22.82 ± 41.65 |
| College Computer Science | -25.63 ± 35.75 |
| Virology | $\infty \pm \infty$ |

Table 8: **ΔIBC Scores by Subject (Qwen-2.5).** Values show the change in IBC from the cascaded LLM framework using the surrogate token probability method, sorted by subject for the (`Qwen-2.5-1.5B` → `Qwen-2.5-7B`) combination, after calibration.

| Subject | ΔIBC (Base → Large) |
|---|---|
| World Religions | 35.95 ± 35.25 |
| Anatomy | 34.10 ± 50.56 |
| Virology | 29.82 ± 77.02 |
| Miscellaneous | 28.80 ± 15.64 |
| High School Psychology | 15.37 ± 14.03 |
| Clinical Knowledge | 14.80 ± 23.85 |
| Prehistory | 14.27 ± 20.97 |
| Marketing | 13.94 ± 27.15 |
| US Foreign Policy | 13.70 ± 34.01 |
| Conceptual Physics | 13.37 ± 22.75 |
| High School Geography | 12.50 ± 29.38 |
| Jurisprudence | 12.22 ± 42.25 |
| Logical Fallacies | 11.77 ± 32.46 |
| Moral Disputes | 11.12 ± 21.71 |
| High School Biology | 7.66 ± 16.00 |
| Management | 6.73 ± 35.36 |
| College Medicine | 5.84 ± 26.08 |
| International Law | 4.88 ± 24.42 |
| High School Microeconomics | 4.38 ± 15.48 |
| Human Aging | 4.16 ± 39.64 |
| High School Macroeconomics | 2.51 ± 14.27 |
| Sociology | 2.05 ± 22.75 |
| Astronomy | 1.97 ± 19.43 |
| Philosophy | 0.88 ± 20.96 |
| Electrical Engineering | 0.33 ± 26.12 |
| Nutrition | 0.31 ± 19.88 |
| College Biology | 0.30 ± 23.37 |
| Abstract Algebra | – |
| Computer Security | -0.26 ± 31.92 |
| All Subjects (Average) | -1.16 ± 2.87 |
| Medical Genetics | -1.39 ± 27.97 |
| Business Ethics | -1.59 ± 35.09 |
| High School Government And Politics | -1.61 ± 16.63 |
| Formal Logic | -2.01 ± 43.78 |
| Professional Psychology | -2.29 ± 13.79 |
| High School Computer Science | -2.58 ± 29.56 |
| Human Sexuality | -3.19 ± 19.56 |
| Security Studies | -4.89 ± 31.41 |
| Public Relations | -5.57 ± 61.18 |
| Econometrics | -5.67 ± 29.45 |
| College Computer Science | -7.06 ± 29.54 |
| High School Statistics | -7.70 ± 15.34 |
| High School Physics | -9.27 ± 25.92 |
| College Physics | -9.43 ± 15.95 |
| Moral Scenarios | -9.63 ± 13.23 |
| Professional Medicine | -10.05 ± 18.46 |
| Elementary Mathematics | -10.43 ± 8.50 |
| Professional Accounting | -10.69 ± 17.82 |
| High School World History | -10.72 ± 19.96 |
| High School US History | -11.81 ± 17.55 |
| High School Mathematics | -12.64 ± 11.41 |
| High School Chemistry | -13.66 ± 17.31 |
| Professional Law | -14.33 ± 10.24 |
| High School European History | -17.91 ± 24.25 |
| College Chemistry | -17.92 ± 38.05 |
| College Mathematics | -17.95 ± 26.48 |
| Machine Learning | -20.64 ± 19.84 |
| Global Facts | -35.32 ± 36.23 |

Table 9: **ΔIBC Scores by Subject (Llama3).** Values show the change in IBC from the cascaded LLM framework using the surrogate token probability method, sorted by subject for the (`Llama3.2-1B` → `Llama3.1-8B`) combination, after calibration.

## C.7 Grid-Search over Threshold Parameters

we perform an additional experiment on the ARC2-Easy dataset with the (Qwen-2.5-1.5B $\rightarrow$ Qwen-2.5-7B) combination. We perform a grid search with every parameter $\theta = \{\phi_{\text{base}}, \xi_{\text{base}}, \xi_{\text{large}}\}$ over $\{0.5, 0.15, 0.25, 0.35, 0.45, 0.55, 0.65, 0.75, 0.85, 0.95\}$. The search grid's time complexity is cubic, $O(n^3)$, and increases with the addition of expert data to the replay buffer, becoming computationally extremely expensive compared to the gradient-based approach. We report our findings in Figure 16. We observe from the results, that the gradient-based approach achieves lower cumulative regret, compared to the grid-search approach, be it on single-model strategies, and also on the cascsaded LLM framework.

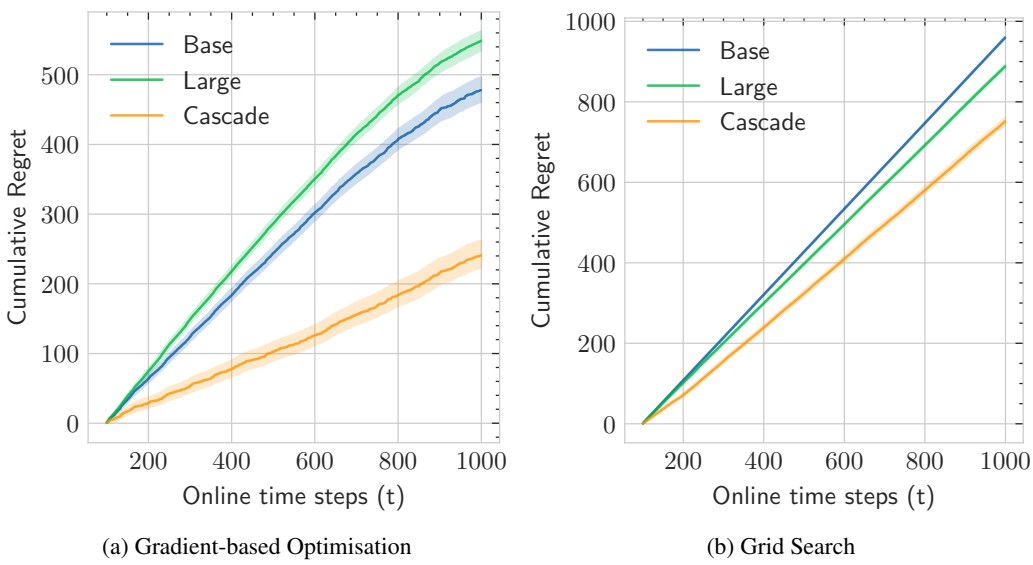

(a) Gradient-based Optimisation  (b) Grid Search

Figure 16: **Gradient-Based vs. Grid-Search**. Online learning performance of the cascaded LLM framework over 1000 samples using the proposed gradient-based (16a) approach against a grid-search (16b) over the threshold parameters $\theta$.

