# OpenReview forum: "Cascaded Language Models for Cost-Effective Human–AI Decision-Making"
_NeurIPS.cc/2025/Conference — NeurIPS 2025 poster_

### Official Review · Reviewer_Jo6m · 2025-06-30

**Clarity:** 2
**Significance:** 2
**Originality:** 2
**Rating:** 3
**Confidence:** 3

**Summary:**

This is an interesting framework that represents an effort to advance the deployment of large language models for human-algorithmic joint decision-making. The effort estimates the cost of doing so and also provides a pathway for on-line deployment and dynamic improvement, evaluating smaller local large language models (LLaMA/Qwen 1B/7B) to provide a proof of concept across several benchmark datasets.

**Questions:**

What is the actual quantifiable benefit for specific use cases?
How would the performance change with larger models?
Are there other benchmarks that would better unify the goals of the authors under this framework?

**Ethical Concerns:**

["NO or VERY MINOR ethics concerns only"]

**Final Justification:**

While I appreciate the limitations that are acknowledged and clarified, I am still unsure if this would be a useful framework without empirical results for larger more commonly available, larger, and updated models. In response to the area chair, my concerns still persist on 1) model level - need to test more complex, widely used, and updated models 2) dataset level - while understandable, the severe limitations of the dataset limit confidence in the findings 3) despite adding a dataset (MMLU), this is supposed to be for clinical applications and 4) I am unconvinced that regret is sufficient since the original work has not been widely adopted or accepted as a viable metric for clinical applications.

**Limitations:**

Yes

**Quality:**

2

**Strengths And Weaknesses:**

This framework considers both cost and performance, estimates uncertainty, and accounts for on-line use of large language models in human-algorithmic interaction for decision-making. The cost-benefit analysis is interesting, and the use of different verification methods anchored by base models of different complexities (albeit both small models).

The use of very small models to answer more complex questions is a major limitation, since the largest open source models are up to 405B parameters. Does the improvement persist with more complex models?The datasets used to test this appear to use multiple choice questions, which are inherently limited in testing for human-AI collaboration, and appear unsuited for on-line use. In the medical domain (MedQA, MedMCQA), there is typically more unstructured question and answer, with variable amounts of information provided throughout the interaction. The authors acknowledge this and then perform the on-line experiment - however, the on-line experiment is poorly described, and the use of Regret should be justified further. How does it actually reflect benefit?

---

> ### Author Rebuttal · Authors · 2025-07-31
>
> We thank the reviewer for their feedback. We address their concerns below:
>
> ---
>
> ### **Q1) Use of Larger Models**
>
> >The use of very small models to answer more complex questions is a major limitation, since the largest open source models are up to 405B parameters. Does the improvement persist with more complex models?
>
> >How would the performance change with larger models?
>
> As our experiments require accessing the logits and, in some instances, also altering the architecture slightly (e.g., Monte Carlo Dropout), we are running all our experiments locally. To this end, we are unfortunately limited to our available compute resources, specifically a single A100 NVIDIA GPU, which makes it impossible to run large models. While larger models lead to better performance, they also vastly increase the number of parameters.
>
> We are sorry that we can not provide more details on this matter.
>
> ---
>
> ### **Q2) Non-QA Datasets**
>
> >The datasets used to test this appear to use multiple choice questions, which are inherently limited in testing for human-AI collaboration, and appear unsuited for on-line use.
>
> Thanks for pointing this out. We tried to run some experiments on the AIME 2022 - 2024 validation set (`AI-MO/aimo-validation-aime`). However, given the limited time during this rebuttal period, we ran into some challenges on this set:
>
> 1. The validation set is only very small, having 30 questions per year a (total of 90 questions).
> This hinders the performance of a good and stable calibration with the Bayesian Logistic Regression.
> 2.  We were having trouble achieving competitive results with available open-source models that would fit our computing resource limitations. We used `Qwen/Qwen2.5-Math-1.5B-Instruct` and `Qwen/Qwen2.5-Math-7B-Instruct` for the cascade of models, where they achieved an accuracy of 8.88% and 14.44%, respectively. These low accuracy values also hinder proper calibration, as the base model should not be trusted solely, and the cascaded system should always abstain.
>
> We apologise for not being able to provide more detailed results on this front at this time; we are working to find a comparable reasoning dataset (e.g. GMS8K), which is not QA for the final version of the manuscript
>
> **ACTION**: Try GSM8K for mathematical reasoning and include it in the main paper, or further acknowledge this shortcoming.
>
> ---
>
> ### **Q3) Unstructured Medical Data**
>
> >In the medical domain (MedQA, MedMCQA), there is typically more unstructured question and answer, with variable amounts of information provided throughout the interaction.
>
> We agree with you that in the medical domain, conversations are more unstructured, and we believe this is a crucial point to address in the future. However, this slightly exceeds the scope of the current paper, as we are interested in the methodologies of a cascaded system and how it can be optimised when deployed online. MedQA and MedMCQA are, nevertheless, both question-and-answer datasets that test specific medical domain knowledge.
>
> ---
>
> ### **Q4) Further Describe the Online Experiment**
>
> >the on-line experiment is poorly described,
>
> We apologise for the brevity of the online‑learning description and will expand it in the revision.  Specifically, the experiment streams 1,000 unseen questions (ARC-Easy, ARC-Challenge, MedQA, MedMCQA) in a *random order* to simulate production traffic.  Before the first query, verification methods are calibrated using the values learned from the 100-sample calibration set; thereafter, we add queries that were marked for abstention and answered by an oracle expert to a replay buffer. This replay buffer is used to perform Adam updates on the differentiable risk (Eq. 2) with a learning rate of 0.05.  Please note that the prediction is made on an unseen query and the regret is calculated on it. If a query is added to the replay buffer, the regret associated with it has already been calculated, and the sample now becomes a training sample; however, it is not further evaluated.  We compare against two baselines assessed on the *same* stream: (i) the single base model, and  (ii) the single large model. We hope this clarifies the matter.
>
> We hope that this clarifies any concerns the reviewer may have had.
>
> ---
>
> ### **Q5) Justification of Regret**
>
> >the use of Regret should be justified further. How does it actually reflect a benefit?
>
> We chose regret as the metric for this experiment, inspired by the work on online decision mediation [4]. More precisely, we believe that regret is a good indicator because:
>
> 1. Regret is just the running sum of our per‑query risk. Because error, compute, and human‑hand‑off are already weighted into the same units, adding them over time tells you the exact “bill” the system has paid.
> 2. A lower regret curve indicates higher benefit, as a combination of abstentions, correct predictions and cost, and we can see it grow over time in a deployed setting.
> 3. Unlike a static accuracy figure, the regret curve illustrates how quickly a policy learns online and whether early mistakes are compensated for later.
>
> Thus, cumulative regret is an additive, monetisable ledger that directly reflects the practical benefit of one routing policy over another.
>
> [4]  Daniel Jarrett, Alihan Hüyük, and Mihaela van der Schaar. Online Decision Mediation. October
> ICLR 2022.
>
> ---
>
> ### **Q6) Other Benchmarks**
>
> >Are there other benchmarks that would better unify the goals of the authors under this framework?
>
> Another benchmark proposed by the reviewers that could be interesting to evaluate is the MMLU benchmark, which tests QA in various domains, including management, foreign relations, and science. Etc.
>
> We have run our experiments and report the following:
> **All MMLU categories**
>
> **$\Delta$IBC**
>
> | **Step** | **Qwen2.5 1.5B → 7B**  | **LLaMA3 1B → 8B** |
> | --- | --- | --- |
> | $\Delta$IBC (STP n=1) | 6.2 ± 5.44 | 5.3 ± 10.57 |
>
> We see that in both cases the $\Delta$IBC is positive. However, it is clearer in the Qwen2.5 cascaded system, where the standard error does not cross zero.
>
> **Online Experiment**
> (While we cannot provide an image, we try our best to give an informative table below)
>
> |  | **Qwen2.5 1.5B → 7B**  |  |  |  | **LLaMA3 1B → 8B** |  |  |
> | --- | --- | --- | --- | --- | --- | --- | --- |
> | **Online Step** | **Cascaded** | **Base** | **Large** |  | **Cascaded** | **Base** | **Large** |
> | **200** | **73.1** | 95.97 | 104.97 |  | **100.46** | 102.92 | 139.61 |
> | **400** | **205.66** | 269.08 | 281.64 |  | **296.33** | 302.75 | 398.17 |
> | **600** | **328.77** | 394.4 | 411.07 |  | **472.75** | 475.32 | 615.78 |
> | **800** | **452.75** | 516.18 | 530.5 |  | 631.86 | **630.86** | 839.49 |
> | **1000** | **589.03** | 649.55 | 665.2 |  | 818.74 | **813.61** | 1042.03 |
>
> We observe that on the combined MMLU dataset, the cascade system still has the lowest cumulative regret after 1000 steps.
>
> **MMLU Subcategories**
>
> Here we also demonstrate some subcategories in an ordered fashion on the **Qwen2.5 1.5B → 7B** cascade, based on $\Delta$IBC
>
> | Subject | $\Delta$IBC ± SE |
> | --- | --- |
> | Computer Security | 113.38 ± 553.75 |
> | High School Psychology | 62.69 ± 41.61 |
> | Sociology | 54.23 ± 77.31 |
> | Public Relations | 47.91 ± 176.82 |
> | … |  |
> | Human Aging | 7.61 ± 47.08 |
> | **All** | **6.20 ± 5.44** |
> | Econometrics | 6.11 ± 42.79 |
> | … |  |
> | High School US History | -21.87 ± 29.37 |
> | Global Facts | -24.72 ± 43.40 |
> | College Physics | -27.22 ± 26.32 |
> | College Chemistry | -29.80 ± 39.77 |
> | Electrical Engineering | -34.56 ± 40.97 |
>
> **ACTION:** Add this new analysis to the central part of the manuscript
>
> [4] Hendrycks, Dan; Burns, Collin; Basart, Steven; Zou, Andy; Mazeika, Mantas; Song, Dawn; Steinhardt, Jacob (2021). "Measuring Massive Multitask Language Understanding". ICLR. arXiv:2009.03300.
>
> ---
>
> We thank the reviewer for their time and hope that we could clarify and address their concerns and hopefully showcase the relevance of our work.

---

> > ### Comment · Reviewer_Jo6m · 2025-08-06
> >
> > I will increase my score to borderline reject (3).
> >
> > I appreciate the efforts the authors have made to think through the challenges and introduce more information regarding their choices. With the new results the authors should be more clear about the very limited impact this may have in the medical domain and the severely constrained environment that limits comparisons to larger models.

---

> > > ### Author Response · Authors · 2025-08-06
> > > **Thank you**
> > >
> > > We thank the reviewer for their time and constructive feedback. Moreover, we appreciate the raised score.
> > >
> > > We will make sure to introduce these limitations.

---

### Official Review · Reviewer_D2qF · 2025-07-01

**Clarity:** 2
**Significance:** 2
**Originality:** 3
**Rating:** 3
**Confidence:** 3

**Summary:**

The paper proposes a three-tier cascade that routes tasks from a lightweight LLM to a larger LLM or a human expert via calibrated confidence-based deferral and abstention thresholds, updating these thresholds online to balance accuracy, cost, and risk. Across four multiple-choice benchmarks (ARC-Easy/Challenge, MedQA, MedMCQA), the cascade cuts inference cost while maintaining or improving accuracy.

**Questions:**

please refer to weakness

**Ethical Concerns:**

["NO or VERY MINOR ethics concerns only"]

**Final Justification:**

I appreciate the authors’ detailed responses, which address some of my concerns. However, there are still several important aspects that should be included in the camera-ready version of the paper. Therefore, I would like to keep my score unchanged at this time. I strongly recommend that the authors incorporate a more comprehensive evaluation in the final version.

**Limitations:**

please refer to weakness

**Quality:**

2

**Strengths And Weaknesses:**

**Strengths:**

1. The exploration of multi-agent collaboration—combining small models, large models, and human experts—is an interesting direction that has the potential to further push the limits of LLM capabilities.
2. The paper is clearly written and easy to follow.

**Weaknesses:**

1. The experimental setup lacks clarity, especially regarding which versions of LLaMA and Qwen were used. It remains unclear whether the improvements from cascading would diminish as base models become stronger. This missing analysis is critical.
2. The evaluation is limited to a few multiple-choice tasks. Key benchmarks like MMLU, MATH, and AIME, which test general and reasoning ability, are missing. It’s important to assess whether the proposed framework is still effective—and whether it can enhance LLMs further—in more challenging scenarios.
3. The paper does not compare against other multi-agent coordination methods. It remains unclear how this cascade framework performs relative to existing autonomous agent collaboration strategies.
4. As shown in Figure 4(c), the cascade fails to outperform a single large model on MedQA, suggesting that its generalization to complex, domain-specific tasks needs further validation.
5. In real-world settings, accurate human feedback can be sparse or noisy, which would make online threshold tuning difficult and limit the applicability of the framework.
6. The overall novelty of the work is relatively limited.

---

> ### Author Rebuttal · Authors · 2025-07-31
>
> We thank the reviewer for their constructive feedback and appreciate that they view this as an interesting problem.
>
> We address the weaknesses above in our responses below
>
> ---
>
> ### **W1) Clarity of Experimental Setup**
>
> >The experimental setup lacks clarity, especially regarding which versions of LLaMA and Qwen were used. It remains unclear whether the improvements from cascading would diminish as base models become stronger. This missing analysis is critical.
>
> Thank you for pointing this out. Apologies for not having specified this clearer in the paper. We use the following models:
> - `meta-llama/Llama-3.2-1B-Instruct`
>
> - `meta-llama/Llama-3.2-3B-Instruct`
>
> - `meta-llama/Llama-3.1-8B-Instruct`
>
> - `Qwen/Qwen2.5-1.5B-Instruct`
>
> - `Qwen/Qwen2.5-3B-Instruct`
>
> - `Qwen/Qwen2.5-7B-Instruct`
>
> As our experiments require accessing the logits and, in some instances, also altering the architecture slightly (e.g., Monte Carlo Dropout), we are running all our experiments locally. To this end, we are unfortunately limited to our available compute resources, specifically a single A100 NVIDIA GPU, which makes it impossible to run large models. While larger models lead to better performance, they also vastly increase the number of parameters.
>
> To address this problem, we have tried an experiment with `Qwen/Qwen2.5-1.5B-Instruct` and `Qwen/Qwen2.5-14B-Instruct` And we report the results here:
>
> **ACTION**: Add a more detailed explanation of the models and experiments in the main paper, as well as in the appendix
>
> ---
>
> ### **W2 & W4) Additional Benchmarks**
>
> >The evaluation is limited to a few multiple-choice tasks. Key benchmarks like MMLU, MATH, and AIME, which test general and reasoning ability, are missing. It’s important to assess whether the proposed framework is still effective—and whether it can enhance LLMs further—in more challenging scenarios.
>
> >As shown in Figure 4(c), the cascade fails to outperform a single large model on MedQA, suggesting that its generalization to complex, domain-specific tasks needs further validation.
>
> Thank you for this comment. We ran some experiments and you can find the results here on **Qwen2.5 1.5B → 7B**:
>
> **All MMLU categories**
>
> Thank you for pointing this out. To this extend we have run our experiments during this short rebuttal time on MMLU [4], to assess the wider variety of the cascaded model systems on questions beyond grade-shool science question, but also including foreign politics, international law, logic, or management
>
> We report the results here:
>
> **All MMLU categories**
>
> **$\Delta$IBC**
>
> | **Step** | **Qwen2.5 1.5B → 7B**  | **LLaMA3 1B → 8B** |
> | --- | --- | --- |
> | $\Delta$IBC (STP n=1) | 6.2 ± 5.44 | 5.3 ± 10.57 |
>
> We see that in both cases the $\Delta$IBC is positive. However, it is clearer in the Qwen2.5 cascaded system, where the standard error does not cross zero.
>
> **Online Experiment**
> (While we cannot provide an image, we try our best to give an informative table below)
>
> |  | **Qwen2.5 1.5B → 7B**  |  |  |  | **LLaMA3 1B → 8B** |  |  |
> | --- | --- | --- | --- | --- | --- | --- | --- |
> | **Online Step** | **Cascaded** | **Base** | **Large** |  | **Cascaded** | **Base** | **Large** |
> | **200** | **73.1** | 95.97 | 104.97 |  | **100.46** | 102.92 | 139.61 |
> | **400** | **205.66** | 269.08 | 281.64 |  | **296.33** | 302.75 | 398.17 |
> | **600** | **328.77** | 394.4 | 411.07 |  | **472.75** | 475.32 | 615.78 |
> | **800** | **452.75** | 516.18 | 530.5 |  | 631.86 | **630.86** | 839.49 |
> | **1000** | **589.03** | 649.55 | 665.2 |  | 818.74 | **813.61** | 1042.03 |
>
> We observe that on the combined MMLU dataset, the cascade system still has the lowest cumulative regret after 1000 steps.
>
> **MMLU Subcategories**
>
> Here we also demonstrate some subcategories in an ordered fashion on the **Qwen2.5 1.5B → 7B** cascade, based on $\Delta$IBC
>
> | Subject | $\Delta$IBC ± SE |
> | --- | --- |
> | Computer Security | 113.38 ± 553.75 |
> | High School Psychology | 62.69 ± 41.61 |
> | Sociology | 54.23 ± 77.31 |
> | Public Relations | 47.91 ± 176.82 |
> | … |  |
> | Human Aging | 7.61 ± 47.08 |
> | **All** | **6.20 ± 5.44** |
> | Econometrics | 6.11 ± 42.79 |
> | … |  |
> | High School US History | -21.87 ± 29.37 |
> | Global Facts | -24.72 ± 43.40 |
> | College Physics | -27.22 ± 26.32 |
> | College Chemistry | -29.80 ± 39.77 |
> | Electrical Engineering | -34.56 ± 40.97 |
>
> As shown above, different domain-specific tasks can vary significantly in how the cascaded model performs in a static setting. We hope that this further analysis sheds more clarity on the effect of the domain-specificity for our framework.
>
> **AIME (Math reasoning):**
>
> We tried to run some experiments on the AIME 2022 - 2024 validation set (`AI-MO/aimo-validation-aime`). However, given the limited time during this rebuttal period, we ran into some challenges on this set:
>
> 1. The validation set is only very small, having 30 questions per year a (total of 90 questions).
> This hinders the performance of a good and stable calibration with the Bayesian Logistic Regression.
> 2.  We were having trouble achieving competitive results with available open-source models that would fit our computing resource limitations. We used `Qwen/Qwen2.5-Math-1.5B-Instruct` and `Qwen/Qwen2.5-Math-7B-Instruct` for the cascade of models, where they achieved an accuracy of 8.88% and 14.44%, respectively. These low accuracy values also hinder proper calibration, as the base model should not be trusted solely, and the cascaded system should always abstain.
>
> We apologise for not being able to provide more detailed results on this front at this time; we are working to find a comparable reasoning dataset (e.g. GMS8K), which is not QA for the final version of the manuscript
>
> **ACTION**: Add these results to the main paper
>
> ---
>
> ### **W5) Noisy human feedback**
>
> >In real-world settings, accurate human feedback can be sparse or noisy, which would make online threshold tuning difficult and limit the applicability of the framework.
>
> We conducted experiments on the datasets, incorporating random noise when returning feedback that is consistent with human error on these tasks.
>
> We show this on some simple additional experiments in the online setting, where we increasingly swap correct to incorrect predictions, when calibrating the model, and how this affects the trajectory on the ARC-Easy dataset with Qwen2.5-1.5B → Qwen2.5-7B
>
> | **Online Step** | **0%**  | **20%** | **50%** | **100%** |
> | --- | --- | --- | --- | --- |
> | **200** | **28.66** | 44.75 | 74.39 | 97.54 |
> | **400** | **77.01** | 127.12 | 204.69 | 303.3 |
> | **600** | **126.12** | 208.42 | 321.77 | 506.24 |
> | **800** | **183.54** | 294.01 | 435.38 | 698.95 |
> | **1000** | **240.68** | 380.16 | 548.74 | 885.77 |
>
> ---
>
> We observe that the cumulative regret in the online setting increases as the number of wrongly labelled data points increases, which calibrates the $\Phi_{\text{base}}$ and $\Phi_{\text{large}}$.
>
> **ACTION**: highlight this point further in the limitations and add this analysis to the manuscript.

---

> > ### Comment · Reviewer_D2qF · 2025-08-08
> >
> > I appreciate the authors’ detailed responses, which address some of my concerns. However, there are still several important aspects that should be included in the camera-ready version of the paper. Therefore, I would like to keep my score unchanged at this time. I strongly recommend that the authors incorporate a more comprehensive evaluation in the final version.

---

> > > ### Author Response · Authors · 2025-08-08
> > >
> > > We thank the reviewer for their response and for taking the time to review.
> > >
> > > Would it be possible to understand in more detail which several important aspects are missing, such that we could address these?

---

### Official Review · Reviewer_zGHi · 2025-07-04

**Clarity:** 3
**Significance:** 4
**Originality:** 3
**Rating:** 5
**Confidence:** 3

**Summary:**

This paper proposes a cascaded framework for human-LLM decision-making by generating predictions with a small base model, deferring to a more capable but costlier model (e.g. more compute budget) when uncertain, and abstaining to a human expert if even the costlier model is uncertain. The authors further show that across multiple interactions, cumulative regret can be minimized by training the deferral and abstention policies continually in an online fashion when feedback is available.

**Questions:**

Kindly address the questions regarding missing details from the "Weaknesses" section.

**Ethical Concerns:**

["NO or VERY MINOR ethics concerns only"]

**Final Justification:**

The authors have adequately addressed the concerns I raised, and I am raising my score to an Accept accordingly.

**Limitations:**

Yes

**Paper Formatting Concerns:**

-

**Quality:**

3

**Strengths And Weaknesses:**

### Strengths

- **Important and well-motivated problem**

- **Well-designed experiments and analyses:** The experimental design seems fairly sound appropriate for the research questions they are trying to answer. The results seem solid, consistent and convincing. Overall, I am convinced that uncertainty-based LLM cascades are a useful paradigm for cost-effective human-AI collaboration.

- **Writing is clear and easy to follow.** The motivation and contributions are very clear. The methodology is fairly easy to understand.



### Weaknesses

- **Missing details:**
    - Calibration errors/Brier scores for the various confidence calibration methods. A further analysis between miscalibration of the conf estimation method and overall accuracy of the cascaded system would be interesting to see.
    - Who are the human experts in your experiments? Does the accuracy in Figures 3 include the correctness of the experts' predictions, when the large model defers to the expert?
    - How is uncertainty estimation calculated? In L113, it is mentioned that uncertainty $\in [0, \infty]$, but in section 4, it is not clear how this is estimated, or whether it is estimated using the same function as the confidence estimation function. In fact, it is not even clear how confidence and uncertainty scores are different.
- **Low ecological validity of tasks:** The selected datasets are not representative of realistic use cases of LLM systems (grade-school science and medical exam questions).

---

> ### Author Rebuttal · Authors · 2025-07-31
>
> We thank the reviewer for their positive feedback. We’re glad the reviewer finds the problem important and our analyses well-designed with promising results.
>
> We address the weaknesses below.
>
> ---
>
> ### **W1) Calibration**
>
> >Calibration errors/Brier scores for the various confidence calibration methods. A further analysis between miscalibration of the conf estimation method and overall accuracy of the cascaded system would be interesting to see.
>
> Thank you very much for this observation. We agree with you. Unfortunately, due to the limited time in this rebuttal period, we prioritised the other experiments and, therefore, did not manage to complete this analysis in time.
>
> We did however at least inspect how different sizes of the calibration impact the $\Delta$IBC score
>
> | **Model** | **Dataset** | **Cal. Size = 50** | **Cal. Size = 100** | **Cal. Size = 200** | **Cal. Size = 500** |
> | --- | --- | --- | --- | --- | --- |
> | **Qwen 2.5 (1.5B–7B)** | ARC2 Easy | 239.5 ± 36.1 | 242.7 ± 36.3 | **273.5 ± 42.7** | 254.9 ± 43.5 |
> |  | ARC2 Challenge | 87.8 ± 24.3 | **89.3 ± 26.2** | 68.7 ± 25.4 | 62.1 ± 28.7 |
> |  | MedQA | -3.0 ± 15.4 | **1.7 ± 16.3** | -7.7 ± 16.6 | -16.6 ± 18.6 |
> |  | MedMCQA | -7.5 ± 9.9 | -10.0 ± 10.1 | -8.0 ± 9.9 | **-5.7 ± 10.4** |
> | **LLaMA 3 (1B–8B)** | ARC2 Easy | 121.9 ± 15.6 | 118.3 ± 15.8 | **122.6 ± 16.5** | 102.8 ± 16.0 |
> |  | ARC2 Challenge | **45.7 ± 13.1** | 38.1 ± 13.7 | 40.3 ± 12.8 | 43.4 ± 15.6 |
> |  | MedQA | **10.1 ± 9.7** | 5.6 ± 9.7 | 5.3 ± 10.0 | 9.8 ± 11.9 |
> |  | MedMCQA | **7.4 ± 9.5** | -3.7 ± 9.2 | 6.7 ± 9.1 | 7.0 ± 9.3 |
>
> From here, we can see that increasing the calibration set does not significantly increase the final $\Delta$IBC score, particularly given the overlap in standard errors.
>
> **ACTION**: Add Brier score analysis to the appendix of the manuscript for better understanding of the calibration.
>
> ---
>
> ### **W2) Information about Experts**
>
> Who are the human experts in your experiments? Does the accuracy in Figures 3 include the correctness of the experts' predictions, when the large model defers to the expert?
>
> First of all, no, the accuracies in Figure 3 do not include the correctness of the expert predictions. This is also not the case in Figure 4.
>
> In our experiments, we do not use human experts directly; instead, we assume the presence of an oracle that returns the correct answer if the cascaded system decides to abstain from making a decision. Therefore, the human experts are indirectly the people who labelled the 4 datasets respectively.
>
> This might be an oversimplification, and we acknowledge this shortcoming. To this end, we conducted experiments on the datasets, incorporating random noise when returning feedback that is consistent with human error on these tasks.
>
> We show this on some simple additional experiments in the online setting, where we increasingly swap correct to incorrect predictions, when calibrating the model, and how this affects the trajectory on the ARC-Easy dataset with Qwen2.5-1.5B → Qwen2.5-7B
>
> | **Online Step** | **0%**  | **20%** | **50%** | **100%** |
> | --- | --- | --- | --- | --- |
> | **200** | **28.66** | 44.75 | 74.39 | 97.54 |
> | **400** | **77.01** | 127.12 | 204.69 | 303.3 |
> | **600** | **126.12** | 208.42 | 321.77 | 506.24 |
> | **800** | **183.54** | 294.01 | 435.38 | 698.95 |
> | **1000** | **240.68** | 380.16 | 548.74 | 885.77 |
>
> We observe that the cumulative regret in the online setting increases as the number of wrongly labelled data points increases, which calibrates the $\Phi_{\text{base}}$ and $\Phi_{\text{large}}$.
>
> **ACTION**: highlight this point further in the limitations and add this analysis to the manuscript.
>
> ---
>
> ### **W3) Explain Uncertainty**
>
> How is uncertainty estimation calculated? In L113, it is mentioned that uncertainty , but in section 4, it is not clear how this is estimated, or whether it is estimated using the same function as the confidence estimation function. In fact, it is not even clear how confidence and uncertainty scores are different.
>
> Thank you for your question. This is a great point that we should highlight more effectively. The model we use to calibrate our verification data is Bayesian logistic regression (BLR). Compared to a standard logistic regression, a BLR returns a posterior predictive distribution, rather than a single point estimate.  This serves us for two purposes: Firstly, the mean of the predictive distribution is used to recalibrate the confidence scores (just like in a standard logistic regression). Secondly, we utilise the spread of the posterior predictive distribution, specifically the standard deviation, to quantify uncertainty, which we then use to determine abstention.
>
> **ACTION**: Make this distinction clearer in the main manuscript
>
> ---
>
> ### **W4)Low ecological validity of tasks**
>
> The selected datasets are not representative of realistic use cases of LLM systems (grade-school science and medical exam questions).
>
> Thank you for pointing this out. To this extend we have run our experiments during this short rebuttal time on MMLU [4], to assess the wider variety of the cascaded model systems on questions beyond grade-shool science question, but also including foreign politics, international law, logic, or management
>
> We report the results here:
>
> **All MMLU categories**
>
> **$\Delta$IBC**
>
> | **Step** | **Qwen2.5 1.5B → 7B**  | **LLaMA3 1B → 8B** |
> | --- | --- | --- |
> | $\Delta$IBC (STP n=1) | 6.2 ± 5.44 | 5.3 ± 10.57 |
>
> We see that in both cases the $\Delta$IBC is positive. However, it is clearer in the Qwen2.5 cascaded system, where the standard error does not cross zero.
>
> **Online Experiment**
> (While we cannot provide an image, we try our best to give an informative table below)
>
> |  | **Qwen2.5 1.5B → 7B**  |  |  |  | **LLaMA3 1B → 8B** |  |  |
> | --- | --- | --- | --- | --- | --- | --- | --- |
> | **Online Step** | **Cascaded** | **Base** | **Large** |  | **Cascaded** | **Base** | **Large** |
> | **200** | **73.1** | 95.97 | 104.97 |  | **100.46** | 102.92 | 139.61 |
> | **400** | **205.66** | 269.08 | 281.64 |  | **296.33** | 302.75 | 398.17 |
> | **600** | **328.77** | 394.4 | 411.07 |  | **472.75** | 475.32 | 615.78 |
> | **800** | **452.75** | 516.18 | 530.5 |  | 631.86 | **630.86** | 839.49 |
> | **1000** | **589.03** | 649.55 | 665.2 |  | 818.74 | **813.61** | 1042.03 |
>
> We observe that on the combined MMLU dataset, the cascade system still has the lowest cumulative regret after 1000 steps.
>
> **MMLU Subcategories**
>
> Here we also demonstrate some subcategories in an ordered fashion on the **Qwen2.5 1.5B → 7B** cascade, based on $\Delta$IBC
>
> | Subject | $\Delta$IBC ± SE |
> | --- | --- |
> | Computer Security | 113.38 ± 553.75 |
> | High School Psychology | 62.69 ± 41.61 |
> | Sociology | 54.23 ± 77.31 |
> | Public Relations | 47.91 ± 176.82 |
> | … |  |
> | Human Aging | 7.61 ± 47.08 |
> | **All** | **6.20 ± 5.44** |
> | Econometrics | 6.11 ± 42.79 |
> | … |  |
> | High School US History | -21.87 ± 29.37 |
> | Global Facts | -24.72 ± 43.40 |
> | College Physics | -27.22 ± 26.32 |
> | College Chemistry | -29.80 ± 39.77 |
> | Electrical Engineering | -34.56 ± 40.97 |
>
> **ACTION:** Add this new analysis in the main part of the manuscript
>
> [4] Hendrycks, Dan; Burns, Collin; Basart, Steven; Zou, Andy; Mazeika, Mantas; Song, Dawn; Steinhardt, Jacob (2021). "Measuring Massive Multitask Language Understanding". ICLR. arXiv:2009.03300.

---

> > ### Comment · Reviewer_zGHi · 2025-08-06
> > **Response to Rebuttal**
> >
> > Thank you for the detailed response.

---

> > > ### Author Response · Authors · 2025-08-06
> > > **Thank you for reviewing**
> > >
> > > We thank the reviewer for their time and constructive input!
> > >
> > > We hope our response could address their concerns.
> > >
> > > Is there anything that we should clarify to improve the quality of our paper?

---

### Official Review · Reviewer_wErX · 2025-07-11

**Clarity:** 3
**Significance:** 2
**Originality:** 2
**Rating:** 4
**Confidence:** 3

**Summary:**

The paper proposes an end-to-end risk-aware cascade for large-language-model (LLM) question answering that (i) routes queries first to a cheap base model, (ii) decides—via a calibrated confidence/uncertainty gate—whether to accept that answer, defer to a large verifier, or abstain to a human, and (iii) learns all routing thresholds online by stochastic gradient descent on a differentiable objective that simultaneously prices accuracy, compute cost, and human escalation.  The key technical pieces are a surrogate-token probability verifier that costs only a single forward pass of the big model and a soft-mask formulation that makes a three-way routing policy differentiable.  Experiments on four multiple-choice QA benchmarks show sizable cost–accuracy gains over either single model and illustrate the limits of the approach on harder medical tasks.

**Questions:**

- Have you run an offline grid-search or Bayesian-optimisation baseline that chooses a single-gate confidence threshold (or a two-gate confidence + uncertainty router) on a validation split, and if so, how does its cost–risk trade-off compare to your differentiable soft-mask policy?
- Do you have empirical data on expert latency, monetary cost, and inter-annotator disagreement for the tasks you study? If so, can you rerun the risk analysis with those measured λₐ values instead of the assumed constants (λₐ = 0.1, λ_c = 10⁻⁵)?
- How sensitive is performance to the size of the calibration set? Could you show ΔIBC curves for 50, 200, and 500 calibration examples?

**Ethical Concerns:**

["NO or VERY MINOR ethics concerns only"]

**Final Justification:**

The author addressed the concern on not using real human experts and robustness of the result given varying calibration size. The requested comparison to simpler grid search is not addressed; it's a more expensive method though. Overall, I think the new results validate the correctness of the paper.

**Limitations:**

yes

**Quality:**

3

**Strengths And Weaknesses:**

Strengths:
- The paper turns the well-trodden two-tier cascade into a principled, self-optimising, and empirically validated framework for multiple-choice QA. Its main innovations—the risk objective, STP verification, and online threshold learning—are theoretically clean and practically useful.
- Breadth of study—two model families, four benchmarks, extensive ablations—gives credible evidence that the cascade can deliver cost savings. Reproducible: code is released, and ablations clarify which components matter.
- The method is drop-in for discrete-choice QA pipelines and shows clear Pareto improvements on science-QA tasks.

Weaknesses；
- No real human-in-the-loop experiment. The “human tier” is an oracle that instantly returns the benchmark label; consequently  - 1）the framework never measures human turnaround time or error; and 2) λ_a (abstention cost) is set by fiat, not from empirical data. This makes all online-learning curves optimistic.
- Dual gates, soft-mask logistics, and multiple verification variants may be over-engineered. Does simpler grid-searched thresholds (or even single-gate routers) obtain similar gains?

Overall, the work is a solid incremental contribution for production-minded QA systems, but further research is required before the approach can be called a general solution for LLM deployment.

---

> ### Author Rebuttal · Authors · 2025-07-31
>
> We thank the reviewer for their constructive feedback. We’re glad the reviewer finds the approach principled, as well as the breadth of experiments and the empirical results promising and effective.
>
> We address their concerns and questions below:
>
> ---
> ### **W1) Noisy Human in the Loop**
> >No real human-in-the-loop experiment.
> >1）the framework never measures human turnaround time or error;
> The reason why we used the Oracle labeller was to isolate the optimisation behaviour of the cascade. Using an oracle, let us measure the *best‑possible* value of deferring to humans. Nevertheless, we acknowledge that this is a strong assumption.
>
> However, let’s have a look at the defined risk from a mathematical perspective, if we assume that we have an imperfect labeller (such as not all labels are correct)
>
> To recall the system risk objective, which is being optimised, is:
>
> $$\mathcal{R}(C)
> = 1-\mathbb{E}[\text{Correct}] + \lambda_c \mathbb{E}[\text{Cost}] + \lambda_a \mathbb{P}[\text{Abstention}]$$
> (eq.12, page 6)
>
> Now, the part where wrong human annotations will have an impact is at
>
> $$\mathbb{E}[\text{Correct}] = \mathbb{E}\bigl[m_{\text{pred1}}\cdot\Phi_{\text{base}}\bigr] +
> \mathbb{E}\bigl[m_{\text{pred2}}\cdot\Phi_{\text{large}}\bigr]$$
>
> more precisely on the calibrated confidence scores $\Phi_{\text{base}}$ and $\Phi_{\text{large}}$. The noisier the feedback is, the more uncalibrated $\Phi_{\text{base}}$ and $\Phi_{\text{large}}$ will become. Therefore, optimisation of the cascaded model will become unreliable.
>
> We show this on some simple additional experiments in the online setting, where we increasingly swap correct to incorrect predictions, when calibrating the model, and how this affects the trajectory on the ARC-Easy dataset with `Qwen2.5-1.5B-Instruct` → `Qwen2.5-7B-Instruct`
>
> | **Online Step** | **0%**  | **20%** | **50%** | **100%** |
> | --- | --- | --- | --- | --- |
> | **200** | **28.66** | 44.75 | 74.39 | 97.54 |
> | **400** | **77.01** | 127.12 | 204.69 | 303.3 |
> | **600** | **126.12** | 208.42 | 321.77 | 506.24 |
> | **800** | **183.54** | 294.01 | 435.38 | 698.95 |
> | **1000** | **240.68** | 380.16 | 548.74 | 885.77 |
>
> We observe that the cumulative regret in the online setting increases as the number of wrongly labelled data points increases, which calibrates the $\Phi_{\text{base}}$ and $\Phi_{\text{large}}$.
>
> **ACTION**: highlight this point further in the limitations and add this analysis to the manuscript.
>
> ---
> ### **W2 & Q2) Abstention Cost**
> >2) λ_a (abstention cost) is set by fiat, not from empirical data. This makes all online-learning curves optimistic.
> >Do you have empirical data on expert latency, monetary cost, and inter-annotator disagreement for the tasks you study?
>
> Thanks for pointing this out. We want to emphasise that this multi-objective problem is highly dependent on the initial assumptions of $\lambda_a$ and $\lambda_c$. For example, setting $\lambda_a=\infty$ and $\lambda_c=0$, the trivial solution is always to choose the best-performing model, while, e.g., $ \ lambda_a=0$ and $\lambda_c=\infty$, the trivial solution is to always choose the cheapest model.
>
> Our choice for $\lambda_a = 0.1$ and $\lambda = 10^{-5}$ originates from Zellinger et al.’s paper [3], where they analyse the multi-objective problem between performance, cost, and abstention, in various combinations of $\lambda_a$ and $\lambda_c$. From their Figure 3, we see that the choices of $\lambda_a = 0.1$ and $\lambda = 10^{-5}$ are sensible initial assumptions for this multi-objective problem, which are worth investigating on various datasets without yielding trivial solutions.
>
> [3] Michael J. Zellinger, Rex Liu, and Matt Thomson. Cost-Saving LLM Cascades with Early Abstention, March 2025. arXiv:2502.09054
> **ACTION** Justify this part better in the manuscript as well as to acknowledge this limitation
>
> ---
> ### **W3) Over-engineered method and grid-search**
> >Dual gates, soft-mask logistics, and multiple verification variants may be over-engineered. Does simpler grid-searched thresholds (or even single-gate routers) obtain similar gains?
> >Have you run an offline grid-search or Bayesian-optimisation baseline that chooses a single-gate confidence threshold (or a two-gate confidence + uncertainty router)
>
> Thank you for pointing this out. First of all, we wish to point out that the exploration of the multiple verification methods is independent of the dual gates and is used to determine which verification is a sensible choice. We agree that a simple threshold grid search might be sufficient. However, a main concern about grid search is that, depending on the granularity of the search, the computational cost can become significantly higher than the differentiable soft-gate. While we have not currently managed to provide the results here, we will introduce a comparison for the camera-ready version
>
> **ACTION** Add comparison to grid search to the appendix of the paper
>
>
> ---
> ### **Q3) Sensitivity to Calibration set size**
> >How sensitive is performance to the size of the calibration set? Could you show ΔIBC curves for 50, 200, and 500 calibration examples?
>
> Thank you for this question. We reran the experiments on 50, 200, and 500 calibration examples, and these are the results:
>
> | **Model** | **Dataset** | **Cal. Size = 50** | **Cal. Size = 100** | **Cal. Size = 200** | **Cal. Size = 500** |
> | --- | --- | --- | --- | --- | --- |
> | **Qwen 2.5 (1.5B–7B)** | ARC2 Easy | 239.5 ± 36.1 | 242.7 ± 36.3 | **273.5 ± 42.7** | 254.9 ± 43.5 |
> |  | ARC2 Challenge | 87.8 ± 24.3 | **89.3 ± 26.2** | 68.7 ± 25.4 | 62.1 ± 28.7 |
> |  | MedQA | -3.0 ± 15.4 | **1.7 ± 16.3** | -7.7 ± 16.6 | -16.6 ± 18.6 |
> |  | MedMCQA | -7.5 ± 9.9 | -10.0 ± 10.1 | -8.0 ± 9.9 | **-5.7 ± 10.4** |
> |  |  | | | |  |
> | **LLaMA 3 (1B–8B)** | ARC2 Easy | 121.9 ± 15.6 | 118.3 ± 15.8 | **122.6 ± 16.5** | 102.8 ± 16.0 |
> |  | ARC2 Challenge | **45.7 ± 13.1** | 38.1 ± 13.7 | 40.3 ± 12.8 | 43.4 ± 15.6 |
> |  | MedQA | **10.1 ± 9.7** | 5.6 ± 9.7 | 5.3 ± 10.0 | 9.8 ± 11.9 |
> |  | MedMCQA | **7.4 ± 9.5** | -3.7 ± 9.2 | 6.7 ± 9.1 | 7.0 ± 9.3 |
>
> Generally, it appears that the calibration size has little influence on these magnitudes. If we examine the standard error across the various sizes, none of them is significantly better than the others. The only thing that we noted were with a too small calibration set, we would have more diverging chains in the Bayesian Logistic Regression sampling.
>
> **ACTION**: Include this analysis in the appendix of the paper.

---

### Note · Authors · 2025-08-12

We are grateful to the reviewers for their insightful feedback. There is a broad consensus that the problem is important and our approach is effective.

- wErX: *“The method is drop-in for discrete-choice QA pipelines and shows clear Pareto improvements on science-QA tasks.”*
- zGHi: *“The results seem solid, consistent and convincing. I am convinced that uncertainty-based LLM cascades are a useful paradigm for cost-effective human-AI collaboration.”*
- D2qF: *“The exploration of multi-agent collaboration […] is an interesting direction that has the potential to push the limits of LLM capabilities further.”*
- Jo6m: *“This is an interesting framework that represents an effort to advance the deployment of large language models for human-algorithmic joint decision-making”*

Concerns focused on missing baselines beyond grade-school science and medical exams, as well as our assumption of perfect human experts.

## Additional Comparison on MMLU
In our responses, we extended our evaluation to the MMLU benchmark, covering 57 subjects from complex STEM to international law, nutrition, and religion. Results are consistent with previous benchmarks on the full test set. We also analysed domain-specific variation, showing that performance gains from our cascaded framework over single-model usage can depend on the subject area.

## Imperfect Human Experts
In the original manuscript, we assumed perfect experts, which we acknowledge can be unrealistic. We ran follow-up experiments simulating imperfect experts by flipping labels in the calibration set to model error rates. As expected, higher human error rates increased cumulative risk in the online decision-making framework, reinforcing the importance of robust human–AI interaction.

## Proposed Revisions
- The inclusion of the MMLU benchmark and the analysis of the performance on the individual subjects
- Including the analysis of having imperfect human experts in the online decision-making setting
- Further clarifying the experiential details.

We  believe that these **proposed changes do not require major changes to our original submission** but rather strengthen the original contributions. We thank the reviewers for their help in enhancing our work.

## Note to AC
We would also like to bring to the AC's attention the **commitment of reviewer Jo6m to raise the score** thanks to the addressed concerns, but as of yet, we can still see the original unchanged score. Potentially, it was forgotten to be updated. Thank you.

---

### Decision · Program_Chairs · 2025-09-17

**Decision:**

Accept (poster)

**Comment:**

This paper presents a cascaded LLM framework for cost-effective human-AI decision-making that routes queries through a multiple-tier system comprising a base model, a larger and higher-capability verifier model, and human experts. The system optimizes a multi-objective risk function designed to balance accuracy, computational cost, and human intervention cost. Experiments across multiple benchmarks (including MMLU, newly added) demonstrate consistent cost-accuracy improvements over single-model baselines.

All reviewers acknowledge this addresses an important practical challenge in LLM deployment, i.e. balancing performance, cost, and human oversight. The framework is methodologically principled and combined well-established techniques in a novel cascade architecture with gradient-based optimization. The results are also consistent across multiple model families and datasets. Reviewers comment also that paper is well-written, and the discussion was helpful in resolving concerns and improving the paper's contributions.

However, the reviewers' scores are split. This is largely due to a few noted weaknesses, among which the most significant in my view are:
- Reviewer D2qF argues (1) the evaluation could be more comprehensive. This is partially mitigated due to the inclusion of MMLU results, (2) limited generality, based on the MedQA results and real-world feedback characteristics, partially mitigated by the noise experiments
- Limits to generality arising from the use of small models (though this is understandable due to computational constraints, and this is not necessarily a direct threat to validity of the core methodology)
- Potentially limited clinical applications, per Jo6m
- Somewhat limited overall novelty
- No comparison to simpler grid-search threshold optimization, though authors acknowledge this would be computationally expensive and commit to including it in the final version (and I strongly recommend that they do so)

On balance, I think this work makes a solid, if modest, contribution to an important problem with technically sound methodology and a compelling empirical validation. So, despite acknowledging some limitations that may constrain its immediate impact, I recommend acceptance.